# Molecular mechanisms of lipid metabolism abnormalities driving sepsis and atrial fibrillation: A Systematic study based on bioinformatics and machine learning

Changze Ou[1,2], Haidong Yu[1], Binbin Chen[1], Huajun Long[2*]

1 Graduate School, Hunan University of Chinese Medicine, Changsha, Hunan Province, China,
2 Department of Emergency, Hunan Provincial Hospital of Integrated Traditional Chinese and Western Medicine (Affiliated Hospital of Hunan Academy of Traditional Chinese Medicine), Changsha, Hunan Province, China

* longhj@hnucm.edu.cn

## Abstract

### Background

Sepsis and atrial fibrillation are complex, life-threatening medical conditions affecting approximately 49 million individuals globally, characterized by exceptionally high mortality rates. Lipid metabolism abnormalities play a critical role in the pathogenesis of these diseases, yet their underlying molecular mechanisms remain incompletely understood.

### Objective

This innovative study systematically investigates the shared molecular mechanisms of lipid metabolism abnormalities in sepsis and atrial fibrillation using advanced bioinformatics and machine learning methodologies. Methods: We retrieved two independent research cohorts from the Gene Expression Omnibus database: sepsis-related datasets and atrial fibrillation-related datasets. A multi-dimensional analytical approach was employed, including differential expression analysis, Weighted Gene Co-expression Network Analysis, machine learning models, immune cell infiltration analysis, and gene set enrichment analysis to comprehensively elucidate the molecular mechanisms of lipid metabolism abnormalities in these diseases.

### Results

Comprehensive analysis identified 13 key candidate genes, with CD81, CKAP4, and DPEP2 emerging as core characteristic genes. Functional enrichment analysis revealed these genes primarily participate in mitochondrial function regulation, complement-coagulation cascade, and cell adhesion molecular pathways. Machine

**Data availability statement:** The datasets analyzed in the current study are available in the Gene Expression Omnibus database (https://www.ncbi.nlm.nih.gov/gds/) with accession codes GSE28750, GSE65682, GSE167363, GSE115574, GSE2240, GSE79768 and GSE41177. All data generated or analysed during this study are included in this published article. All data sets and analyses conducted are accessible upon request by contacting the corresponding author.

**Funding:** This study was supported by the National Traditional Chinese Medicine Expert Inheritance Studio Construction Project (Document No. [2022] 75 from the National Administration of Traditional Chinese Medicine), and the Scientific Research Plan Project of the Health Commission of Hunan Province (Grant No. C202303078160). There was no additional external funding received for this study. The funders had no role in study design, data collection and analysis, decision to publish, or preparation of the manuscript.

**Competing interests:** The authors have declared that no competing interests exist.

learning models demonstrated exceptional diagnostic performance, with area under the curve values of 0.957 for sepsis and 1.000 for atrial fibrillation datasets. Immune cell infiltration analysis unveiled the critical roles of neutrophils and monocytes in disease progression, and revealed the profound impact of lipid metabolism abnormalities on immune regulation.

## Conclusion

We discovered that lipid metabolism abnormalities significantly modulate disease progression by influencing mitochondrial function, inflammatory responses, and cell adhesion pathways. Future research necessitates further clinical validation and functional experiments to explore personalized therapeutic strategies based on lipid reprogramming.

## Introduction

Sepsis is a complex physiological and pathological syndrome triggered by infection. According to the Third International Consensus Definitions Task Force, sepsis is defined as "life-threatening organ dysfunction caused by a dysregulated host response to infection" [1]. Its pathophysiological alterations primarily manifest in four key aspects: endothelial dysfunction, coagulation abnormalities, cellular dysfunction, and cardiovascular dysregulation [2]. Globally, sepsis affects approximately 49 million people annually, with an estimated 11 million deaths attributed to this syndrome, accounting for 19.7% of global mortality [3]. Epidemiological studies in mainland China reveal that 33.6% of intensive care unit (ICU) patients are diagnosed with sepsis, exhibiting a 30-day mortality rate of 29.5%, which is higher than the 24.4% reported in European and American regions [4]. Despite deepening understanding of sepsis pathophysiological mechanisms, effective treatment options remain limited [5]. Atrial fibrillation (AF) is the most common type of cardiac arrhythmia, clinically characterized by palpitations (sensation of rapid, irregular, or pounding heartbeat), fatigue, dizziness, syncope, dyspnea, and stroke [6]. Its incidence is closely associated with age, sex, race/ethnicity, and the presence of comorbid cardiovascular diseases. The global prevalence of AF has significantly increased from 33.5 million in 2010–59 million in 2019 [7]. Current treatments for AF are limited in effectiveness and have a high recurrence rate. Gene therapy, as an emerging treatment modality, may become an important adjunct in future AF management due to its advantages of targeted delivery, tissue specificity, and personalized treatment approach [8]. Research indicates that sepsis is a major risk factor for AF [9], with an incidence of newly diagnosed atrial fibrillation in critically ill patients around 6%, and as high as 33% in patients with sepsis [10]. Meta-analyses show that new-onset AF during sepsis is significantly associated with in-hospital mortality (combined OR: 2.09), post-discharge mortality (combined OR: 2.44), and stroke risk (combined OR: 1.88) [11]. New-onset AF may serve as a marker of sepsis severity, and its underlying mechanisms may be closely related to myocardial necrosis and fibrosis induced by severe inflammation [12].

In the pathological processes of sepsis and AF, abnormalities in lipid metabolism play a critical regulatory role. Their complex pathophysiological mechanisms not only reflect disease states but may also serve as active regulatory mechanisms in disease progression. In sepsis, lipid metabolism disorders present as multi-layered abnormalities, including impaired fatty acid oxidation, dysregulation of lipoprotein levels, and imbalance of inflammatory mediators [13]. In the early stages, enhanced lipolysis in adipose tissue releases large amounts of free fatty acids and glycerol into the bloodstream; however, due to decreased hepatic PPARα expression, fatty acid metabolism is hindered, leading to lipid accumulation and lipotoxic damage in hepatic and renal tissues [14]. High-density lipoprotein (HDL) plays a key role in immune regulation during this process, and its significant decline not only reflects the severity of the disease but also directly impacts inflammatory responses and endothelial function [15]. Similarly, the occurrence of AF is closely related to abnormalities in lipid metabolism, particularly in the context of metabolic syndrome and obesity. Dyslipidemia, characterized by changes in low-density lipoprotein cholesterol (LDL-C), high-density lipoprotein cholesterol (HDL-C), and apolipoprotein A-I (ApoA-I) levels, presents a complex association with AF risk [16]. Very low-density lipoprotein (VLDL) is considered a key factor in the susceptibility to AF among patients with metabolic syndrome, as its accumulation can lead to structural remodeling and functional decline of the atria. VLDL promotes excessive lipid accumulation and apoptosis in atrial cells, further increasing the risk of AF [17]. Lipid metabolism disorders caused by obesity also accelerate the progression of AF by affecting glucose metabolism and lipid accumulation in atrial myocytes, thereby altering the electrophysiological properties of the atria [18]. A common feature of both diseases is the amplification of oxidative stress and inflammatory responses due to lipid metabolism abnormalities [19]. Increased levels of lipid peroxidation products, such as malondialdehyde and 4-hydroxynonenal, exacerbate cellular damage, while changes in the concentrations of signaling molecules like ceramides and lysophosphatidylcholine are closely related to organ dysfunction. Notably, lipid metabolism disorders are not merely passive outcomes; they may actively regulate disease progression. The levels of free fatty acids correlate positively with organ function scores, and dynamic changes in lipid profiles may serve as important biomarkers for disease progression and prognosis [20]. Therefore, targeted modulation of lipid metabolism-related pathways may provide new research directions and potential intervention strategies for the early diagnosis, prevention, and treatment of sepsis and AF.

With the rapid development of bioinformatics and machine learning technologies, unprecedented technical platforms have emerged for in-depth research on the molecular mechanisms of diseases and potential biomarkers. This study systematically integrates bioinformatics and machine learning methods to analyze datasets related to sepsis and AF from the Gene Expression Omnibus (GEO), focusing particularly on the dynamic changes in lipid metabolism. The research employs multidimensional approaches, including the identification of differentially expressed genes related to disease and metabolism, immune cell infiltration analysis, and multilayer enrichment analysis to investigate the crucial role of lipid metabolism in the pathogenesis of sepsis and AF. Notably, abnormalities in lipid metabolism are of significant importance in the occurrence and development of both sepsis and AF, as lipid metabolism disturbances induced by sepsis may lead to atrial remodeling, thereby increasing the risk of AF. Therefore, the innovation of this study lies in systematically exploring the relationship between sepsis and AF, emphasizing the critical role of lipid metabolism in this pathological process. This research provides new insights into further exploration of the shared pathogenesis of the two diseases and their therapeutic strategies, particularly in optimizing interventions related to lipid metabolism.

## Materials and methods

### Collection and organization of research datasets

This study systematically obtained two independent research cohorts from the GEO: sepsis-related datasets (GSE28750, GSE65682, and GSE167363, including 71 patients and 64 controls) and AF-related datasets (GSE115574, GSE2240, GSE79768, and GSE41177, including 84 patients and 69 controls). Among these, GSE28750, GSE115574, GSE2240, and GSE79768 were used as the training set, while the GSE167363 dataset was utilized for single-cell analysis, and GSE65682 and GSE41177 served as independent validation cohorts (Table 1). The training set was exclusively used

**Table 1. Dataset content.**

| GSE series | Disease | Samples | Platform | Tissue | Dataset Category |
|---|---|---|---|---|---|
| GSE28750 | SEPSIS | 10 patients, 20 healthy controls | GPL570 | Blood | Training set |
| GSE65682 | SEPSIS | 750 patients, 42 healthy controls | GPL13667 | Blood | Validation set |
| GSE167363 | SEPSIS | 10 patients, 2 healthy controls | GPL24676 | Blood | Single-cell analysis set |
| GSE2240 | AF | 10 patients, 20 healthy controls | GPL96 | Right atrial appendage | Training set |
| GSE79768 | AF | 14 patients, 12 healthy controls | GPL570 | Human heart | Training set |
| GSE115574 | AF | 28 patients, 31 healthy controls | GPL570 | Atrium | Training set |
| GSE41177 | AF | 14 patients, 12 healthy controls | GPL570 | Left atrial appendage | Validation set |

for differential analysis, model construction, and parameter optimization, while the validation set was completely isolated throughout the process and only employed to verify the efficacy of core genes after model completion, ensuring full independence in their applications. Differences across detection platforms were corrected solely within the training set, further safeguarding the independence of the two datasets to avoid overfitting. The complete analysis workflow is shown in S1 Fig.

## Batch-corrected integration of AF datasets

Raw gene expression matrices of AF datasets were read, with their first column uniformly renamed "geneSymbol" for annotation consistency. Datasets were merged using this column to generate a raw combined matrix, saved in an automatically created timestamped output folder. Filenames served as batch identifiers, and a batch vector was constructed by repeating each identifier according to dataset sample sizes, with consistency verified between vector length and sample columns. The expression matrix was converted to a numeric matrix; batch effects were corrected using limma's removeBatchEffect to eliminate systematic biases while preserving biological differences, with the corrected matrix retained. Normalization involved data scaling via scale.=TRUE during PCA. Correction effectiveness was validated by boxplots (comparing pre- and post-correction expression distributions) and advanced PCA plots (showing batch mixing). A sample-batch table, pre/post-correction PCA score tables, and combined boxplot-PCA figures were generated.

## Differential analysis of gene expression data

To systematically process the gene expression data, bioinformatics analysis tools in R, including limma, ggpubr, pROC, ggplot2, dplyr, and pheatmap, were employed. First, the raw expression matrix of the AF data was log2 transformed and normalized to generate standardized data files. Subsequently, principal component analysis was conducted to assess the differences in sample distribution before and after batch effect correction, with results displayed in scatter plots. Considering the expression characteristics at different stages of disease progression and disease-specific transcriptomic features—sepsis, as a disease driven by acute inflammatory storms, exhibits drastic fluctuations in gene expression, and a threshold of $|logFC| > 1$ can effectively filter noise and focus on high-magnitude inflammation-related signals; in contrast, atrial fibrillation is characterized by chronic, low-magnitude regulatory imbalances, and a threshold of $|logFC| > 0$ helps avoid missing atrial fibrillation-related molecular events with clear biological significance despite low expression differences the differential expression threshold was set: $|logFC| > 1$ for the sepsis group and $|logFC| > 0$ for the AF group, combined with a corrected P value $< 0.05$ as the statistical significance criterion. Differential expression analysis was performed using the limma package to identify significant differentially expressed genes that met the logFC and adjusted P value thresholds, and lists of all differentially expressed genes and significant genes were saved. The expression patterns of significant differentially expressed genes were visualized using heatmaps, while volcano plots illustrated the overall distribution of differential genes.

## Weighted gene co-expression network analysis (WGCNA)

The expression matrix of the GSE28750 dataset underwent quality control, including normalization, low-variance gene filtering, and outlier sample removal, to establish a high-quality expression profile. A scale-free co-expression network was constructed using a soft-thresholding approach, quantifying the co-expression similarity between genes through the topological overlap matrix (TOM). The dynamic tree-cutting algorithm identified gene modules with biological functional coherence, and module eigengenes were further correlated with clinical traits (normal/disease status) to ultimately select key modules significantly associated with the disease. By jointly analyzing gene significance (GS) and module membership (MM), the most strongly associated driver genes within the modules were identified, providing candidate targets for exploring the molecular mechanisms of sepsis-related atrial fibrillation.

## Intersection of disease-metabolism genes

After obtaining the differential genes for sepsis and AF through differential analysis and WGCNA, we conducted a systematic search in the Gene Cards database (https://www.genecards.org/) using "lipid metabolism" as a keyword, resulting in 17,095 related genes. The search results were sorted in descending order by score, and the top 4,000 lipid metabolism-related genes were selected. Among the retrieved lipid metabolism-related genes, a large proportion rely solely on single in silico predictions without wet-lab validation, resulting in low credibility of their association with lipid metabolism. Screening the top 4,000 genes with the highest GeneCards scores (a metric reflecting the strength of experimental evidence for gene-lipid metabolism associations) enables prioritization of core lipid metabolism genes with sufficient experimental support, effectively controlling feature dimensionality, preventing model overfitting, and laying a robust foundation for subsequent analyses. Finally, Venn diagrams were utilized to analyze the intersection of the differential genes from both disease groups with the lipid metabolism gene set, visually presenting the distribution of common genes.

## Pathway and functional enrichment analysis

Using R language tools such as clusterProfiler, org.Hs.e.g.,db, enrichplot, GSEABase, DOSE, and ggplot2, the HGNC symbols of the target genes were first standardized and mapped to Entrez IDs, with a significance threshold set at $P < 0.05$ to ensure statistical reliability. Based on this, Kyoto Encyclopedia of Genes and Genomes (KEGG) pathway enrichment analysis was systematically conducted, along with annotation analysis targeting the three major aspects of Gene Ontology—biological process (BP), molecular function (MF), and cellular component (CC)—and further integrated disease ontology (Disease Ontology, DO) information to elucidate potential pathological associations. For the significantly enriched entries selected through the testing criteria, bar plots and bubble plots were utilized to visualize pathway enrichment intensity, gene proportions, and distribution characteristics, thereby intuitively presenting the functional connections between candidate signaling pathways and disease phenotypes.

## Lasso regression analysis

In this study, LASSO regression (Least Absolute Shrinkage and Selection Operator) was employed to screen for key genes associated with grouping, with a standardized workflow implemented throughout to ensure analytical reliability and result reproducibility. During the data preprocessing phase, gene expression data were imported, transposed into a "sample-feature" format, and grouping information (control group and experimental group) was extracted from the row names. A random seed (123) was set to ensure consistent algorithm iteration, and a binomial distribution LASSO model (alpha = 1) was constructed based on the glmnet package. 10-fold cross-validation (via the cv.glmnet function, type.measure = 'deviance') was used to determine the optimal regularization parameter λ, where λ_min represents the threshold that minimizes cross-validation deviance. Genes with non-zero coefficients at this threshold were selected as candidate biomarkers, and the gene list along with coefficient information was exported. Cross-validation curves (illustrating the relationship

between λ and deviance) and LASSO coefficient path plots (annotating the variation trend of gene coefficients with λ) were generated using the ggplot2 and ggrepel packages, intuitively presenting the feature screening process and results.

### Random forest classification model construction

In this study, the random forest algorithm was adopted to screen for key genes, and the analytical workflow strictly followed standardized principles to ensure result reliability. Based on the preprocessed gene expression matrix, which was transposed into a "sample-feature" format, grouping information was parsed from sample names. A random seed (12345) was set to ensure reproducible model iteration, and the randomForest package was invoked to construct a binary classification model. Initially, 500 decision trees were used to evaluate the variation trend of model error rate with the number of trees, and the optimal number of decision trees corresponding to the minimum error rate was determined accordingly. After retraining the model with optimized parameters, the Gini index (MeanDecreaseGini) was used to quantify the importance of each gene in disease phenotype discrimination. The top 10 candidate genes with high contribution were selected and the results were exported. Meanwhile, ggplot2 was used to plot the error rate curve (showing the variation of model stability with the number of trees) and the gene importance bubble plot (presenting the relative weight and gradient distribution of the top 30 genes), intuitively demonstrating model performance and the contribution differences of key driver genes.

### Support vector machine feature selection analysis

In this study, a linear Support Vector Machine (SVM) with Recursive Feature Elimination (SVM-RFE) model was constructed using the R package e1071 for feature gene screening. For data preprocessing, the gene expression matrix was imported, transposed into a "sample-feature" format, and group information embedded in row names was extracted and converted to a factor type. Non-grouping features underwent z-score normalization to eliminate dimensional interference, and a random seed (12345) was set to ensure consistent algorithm iteration. A multiple SVM-RFE strategy (k = 10) was used for feature screening: a linear kernel SVM built the classification model, with feature contribution to disease discrimination quantified by the sum of squared feature weights; a dynamic elimination mechanism was integrated, removing 50% of low-importance features at once when surviving features exceeded the threshold (halve.above=50) and one feature sequentially when below the threshold. To avoid data leakage and lock the validation pipeline, a nested 10-fold cross-validation framework was adopted: the inner loop used training subsets of each fold for SVM-RFE feature ranking, with the tune function performing grid search within the preset hyperparameter space (gamma: 2^(−12:0), cost: 2^(−6:6)) to select optimal parameters; the outer loop evaluated generalization error on test subsets not involved in feature screening to ensure independent performance assessment. The optimal feature set was determined by calculating the average feature ranking across 10 folds. Generalization error data of the top 30 features were extracted, with the "no-information rate" (random classification error based on sample group proportion) as the baseline, and generalization error curves (marking the number of features corresponding to minimum error) and classification accuracy curves (accuracy = 1 − generalization error) were plotted in a combined layout. Feature screening and parameter optimization were exclusively performed using the training set, while the external validation set was sequestered throughout and only used for final model testing to strictly ensure data independence.

### Validation and evaluation of feature genes

First, the expression matrices of sepsis and AF were loaded, and the grouping labels for disease and healthy controls were extracted based on sample names (encoded as 0/1). The feature gene list was then read, and the corresponding expression profiles were extracted. A two-sample t-test was performed for each gene by group to obtain significance and standard error, while group means were calculated to assess expression trends. In terms of diagnostic performance

evaluation, gene expression levels were used as continuous predictors, and clinical grouping served as the outcome. The pROC package was utilized to construct receiver operating characteristic (ROC) curves and estimate the area under the curve (AUC) and its 95% confidence interval through bootstrapping. Visualization relied on ggplot2 to generate box plots (including significance annotations), density distribution plots, and ROC curve plots to intuitively display expression differences and diagnostic discrimination capabilities between controls and cases. The clinical discriminative value of the feature genes was then evaluated using external datasets GSE65682 and GSE41177. Finally, a nomogram was generated using the rms package to quantify individual disease probabilities, with 1,000 bootstrap samples used to calibrate the predicted probabilities against actual incidence rates. Based on this, the model output probabilities were mapped to net benefit curves at different thresholds, completing decision curve analysis to assess the added value of this multi-gene model in clinical decision-making, ultimately outputting the nomogram, calibration curve, and DCA curve.

## Immune infiltration analysis and its relationship with feature genes

Immune cell infiltration and feature gene association analyses were performed using a modified CIBERSORT algorithm, with the LM22 standard immune cell signature matrix (containing specific gene markers for 22 human immune cell subtypes) as the reference, implemented via the e1071 and preprocessCore packages in R. Data preprocessing included: quality assessment of mixed transcriptome data ($2^{\wedge}$ inverse log transformation performed when the maximum value < 50); quantile normalization using the normalize.quantiles function to eliminate platform differences; batch effect correction with "dataset source + detection platform" as covariates; and z-score standardization for genes overlapping between the reference matrix and mixed data. Immune cell abundance estimation employed a linear kernel ν-SVR model (iterative modeling with ν = 0.25, 0.50, 0.75), with the optimal model selected by minimizing root mean square error (RMSE), and weight vectors normalized to relative abundance. An empirical null distribution was generated via 1000 permutation tests to calculate Pearson correlation coefficients and empirical p-values, and credible samples were filtered by RMSE < 0.1 and p < 0.05. Group-wise analyses were conducted by grouping samples based on suffixes of sample names; wide-format data were converted to long-format for visualization: stacked bar charts for immune cell composition; box plots overlaid with scatter plots combined with Wilcoxon rank-sum tests (Benjamini-Hochberg method for multiple test correction) to show abundance differences, with mean, median, and interquartile range labeled; ridge density plots overlaid with 95% confidence intervals to present subgroup distributions. Spearman tests were used for association analyses between feature genes and immune cells; significant associations were defined by Benjamini-Hochberg corrected p < 0.05, and associations were categorized as weak (|r| < 0.2), moderate (0.2–0.7), and strong (|r| ≥ 0.7) based on absolute correlation coefficients. Correlation heatmaps (with significant association pairs labeled) and scatter plots were generated using the linkET package to visualize association trends, with r and corrected p-values annotated.

## GSEA pathway enrichment analysis of feature genes

In the sepsis and AF datasets, we calculated the median expression values of the feature genes. Based on whether the expression levels of these genes exceeded the median in each sample, we divided the samples into high-expression and low-expression groups and calculated the average expression levels of these two groups to derive log-fold changes. Subsequently, gene set enrichment analysis (GSEA) functions were utilized to analyze the data, extracting five significant pathways (p < 0.05) and plotting GSEA graphs for upregulated and downregulated pathways to visually present the enrichment status of each pathway and their roles in different expression groups.

## ceRNA network prediction

We analyzed three datasets: miRDB, TargetScan, and miRanda, ensuring consistency in all feature genes by printing column names. Each dataset's column names were standardized to "miRNA" and "Gene." A flag column was added to

each dataset to indicate whether the miRNA was present in that database. The three datasets were fully merged based on genes and miRNAs, filling missing values with 0 and calculating the sum for each entry. Subsequently, miRNAs related to key genes were filtered, ensuring only entries present in all databases were retained. Finally, the filtered entries were used to find all lncRNAs associated with these miRNAs using the "sponge_scan" dataset.

## Single-cell transcriptome analysis

Single-cell transcriptome analysis was performed using a standardized workflow constructed with specialized R packages including Seurat, SingleR, limma, and ggplot2. Raw 10X Genomics data of each sample—comprising gene expression matrices, gene names, and cell barcodes—were read via the Read10X function. An error-capturing mechanism was implemented to handle abnormal samples; valid sample data were merged using do.call(cbind, counts) with validity verified, and the analysis was terminated if no valid data were available. A Seurat object was created from the merged data. Initial filtering was set to retain genes expressed in at least 5 cells and cells with at least 200 detected genes. Secondary quality control was conducted to keep cells with nFeature_RNA > 300 and mitochondrial gene (identified by the "MT-" prefix) proportion < 20%. The number of cells before and after filtering was recorded, and the analysis was terminated if fewer than 10 cells remained. Concurrently, violin plots with the theme_classic theme were generated to display the distribution of nFeature_RNA, nCount_RNA, and percent.mito; scatter plots were used to visualize the correlations between nCount_RNA and mitochondrial proportion, as well as between nCount_RNA and nFeature_RNA, with point size set to 1.5. In the preprocessing stage, data normalization was performed using the LogNormalize method with a scaling factor of 10,000. The vst method was applied to select 1,500 highly variable genes, which were visualized. After data standardization, principal component analysis (PCA) was conducted for dimensionality reduction, with 22 principal components (PCs) set. Gene loading plots (including 25 feature genes) and heatmaps (including 400 cells) for the first 3 PCs were visualized; variance contribution rates were calculated, and the optimal number of PCs was determined based on an 80% cumulative variance threshold. A K-nearest neighbor graph was constructed using the first 15 PCs, and clustering was performed with the Louvain algorithm at a resolution of 0.6. Uniform Manifold Approximation and Projection (UMAP) dimensionality reduction (with n.neighbors = 30 and min.dist = 0.3) was applied to generate a clustering plot with the Set3 color palette, and a cell-cluster correspondence table was exported. Automatic cluster annotation was achieved via SingleR using the HumanPrimaryCellAtlasData reference library from the celldex package, and annotations were written back to the Seurat object; a cluster-cell type correspondence table was exported. Positive differentially expressed genes were screened using the FindAllMarkers function (based on the Wilcoxon rank-sum test with Benjamini-Hochberg correction) under the criteria of avg_log2FC > 1, p_val_adj < 0.05, and expression in at least 20% of cells. A table of significant markers was exported, and a heatmap of the top 10 markers per cluster was generated. Simultaneously, the proportion of cell types in each sample was calculated, and stacked bar charts (Set3 color palette), marker count bar charts, and Alluvial flow charts of the top 5 markers were plotted; cell count tables, proportion matrices, and full-cell annotation tables were output. For core genes, their presence was verified, followed by the generation of UMAP expression distribution plots and violin plots grouped by cell type/cluster. The average expression levels of genes in each cell type/cluster and single-cell expression data were exported. Combined UMAP-bar plots were constructed, with clusters and cell types labeled; bar plots displayed the average expression levels of clusters with cell types annotated.

## Potential drug prediction

In this study, key feature genes identified from the machine learning models were utilized as input data for drug prediction. Gene-drug association analysis was conducted using the DSigDB drug database on the Enrichr online platform (https://maayanlab.cloud/Enrichr/). The drugs were ranked based on the strength and frequency of their interactions with the genes. Additionally, factors such as the mechanism of action, safety, and potential therapeutic value of the drugs were comprehensively considered to select candidate drugs with the highest therapeutic potential.

## Results

### Processing of disease datasets and identification of differential genes

Through differential expression gene analysis of transcriptomic data from sepsis patients, we identified 825 significantly altered genes. Among these, 379 (45.9%) were downregulated, and 446 (54.1%) were upregulated, reflecting a significant change in the transcriptomic expression profile under disease conditions (Fig 1A, 1B). Based on WGCNA, we identified three key modules associated with sepsis, with the MEturquoise module showing the most significant correlation with sepsis, containing a total of 1,271 genes (Fig 1C, 1D). After batch effect correction (Fig 1E, 1F), we further identified 1,256 differentially expressed genes in the atrial fibrillation cohort, of which 527 (42.0%) were downregulated and 729 (58.0%) were upregulated, revealing abnormal gene expression characteristics in the atrial fibrillation state (Fig 1G, 1H). Through systematic retrieval from the Gene Cards database, we obtained a gene set of 4,000 related to lipid metabolism. Cross-analysis ultimately identified 13 core candidate genes common to sepsis, atrial fibrillation, and lipid metabolism, including LDHB, CD81, PFKFB2, G0S2, GLRX, SLC22A4, CKAP4, CXCR4, DPEP2, CLU, CDK4, RORA, and BCL11B (Fig 1I, 1J). The expression pattern changes of these genes provide key insights into the potential molecular associations between sepsis and atrial fibrillation in the lipid metabolism pathway, opening new research directions for understanding the pathogenesis of these two complex diseases and their possible shared molecular regulatory networks.

### Pathway and functional enrichment analysis

KEGG pathway analysis indicated that these intersecting genes were primarily enriched in the following biological processes: Hepatitis C, Human cytomegalovirus infection, Propanoate metabolism, and Fructose and mannose metabolism (Fig 2A). Disease ontology enrichment analysis revealed that these intersecting genes were closely related to diseases such as prostatic hypertrophy, prostate disease, primary immunodeficiency disease, and pancreatic adenocarcinoma (Fig 2B). Furthermore, Gene Ontology functional annotation analysis showed that the biological processes (BP) mainly involved the regulation of mitochondrion organization, regulation of the release of cytochrome c from mitochondria, and negative regulation of mitochondrion organization; cellular components (CC) primarily included secretory granule lumen, cytoplasmic vesicle lumen, and vesicle lumen; and molecular functions (MF) mainly involved 14-3-3 protein binding, cholesterol binding, and copper ion binding. These analyses reveal the potential biological significance of these intersecting genes in various diseases, providing an important foundation for further research into the molecular mechanisms of sepsis and atrial fibrillation (Fig 2C, 2D).

### Machine learning selection of feature genes

This study first applied Lasso regression to select feature genes for sepsis. In the Lasso regression model, the λ value reflects the strength of regularization: a larger λ value increases the penalty on coefficients, compressing most regression coefficients close to zero, indicating fewer variables included in the model; as λ decreases, the penalty strength reduces, and more variable coefficients begin to deviate from zero, meaning these variables are gradually incorporated into the model (Fig 3A). Under high λ conditions, we observed that the genes GLPX, CKAP4, CXCR4, and CLU maintained significant positive values, suggesting that these four genes may be key predictors of sepsis. Each red point in the figure represents the bias at different λ values, while the gray error bars indicate statistical uncertainty. The core objective of the study is to find a point that minimizes model bias while keeping λ within a reasonable range to avoid overfitting. The λ value corresponding to the left dashed line represents the optimal predictive performance point of the model, where the model error is minimized with seven genes included (Fig 3B). In the selection of feature genes for atrial fibrillation, LDHB, DPEP2, BCL11B, CKAP4, RORA, PFKFB2, and CXCR4 were identified as potentially important predictors, with the model error being lowest when the number of genes reached eleven (Fig 3C, 3D). Subsequently, we employed a random forest model to identify feature genes associated with the disease by calculating gene importance scores. As the number of decision

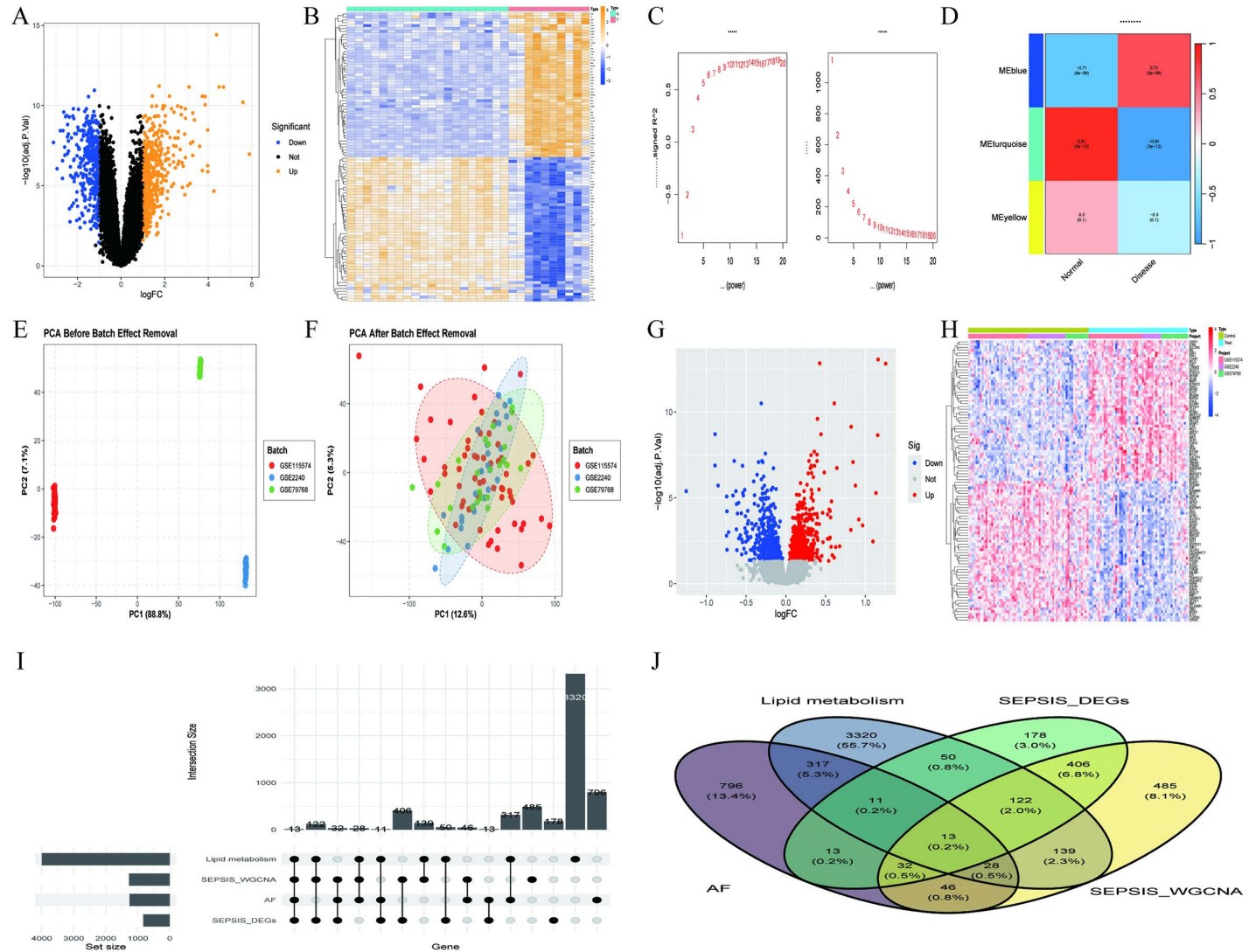

**Fig 1. Analysis of differentially expressed genes in sepsis, atrial fibrillation, and lipid metabolism. (A):** Volcano plot of differentially expressed genes in the GSE28750 dataset, with yellow indicating upregulated genes and blue indicating downregulated genes. **(B):** Heatmap of the top fifty differentially expressed genes in the GSE28750 dataset, where yellow represents upregulation and blue represents downregulation. **(C):** Optimal soft-thresholding plot from the WGCNA of the GSE28750 dataset. **(D):** Module correlation plot from the WGCNA analysis of the GSE28750 dataset. **(E):** PCA distribution plot of the atrial fibrillation dataset before batch effect correction. **(F):** PCA distribution plot of the atrial fibrillation dataset after batch effect correction. **(G):** Volcano plot of differential analysis in the atrial fibrillation dataset, with red indicating upregulated genes and blue indicating downregulated genes. **(H):** Heatmap of the top fifty differentially expressed genes in the atrial fibrillation dataset, where yellow represents upregulation and blue represents downregulation. **(I):** Bar chart illustrating the intersection of differentially expressed genes among sepsis, atrial fibrillation, and lipid metabolism datasets. **(J):** Venn diagram showing the intersection of differentially expressed genes related to sepsis, atrial fibrillation, and lipid metabolism.

trees in the random forest increased, the error rate curve gradually stabilized, indicating that the model's predictive results became more stable and reliable with an increasing number of trees. Typically, the number of trees at which the error rate reaches its lowest point (the lowest point of the black solid line in the figure) is considered the optimal number of decision trees, achieving the best balance between model complexity and accuracy. The gene importance bubble plot generated by the random forest algorithm visually displays the genes with the highest importance in the model (Fig 3E-3H). Next, we

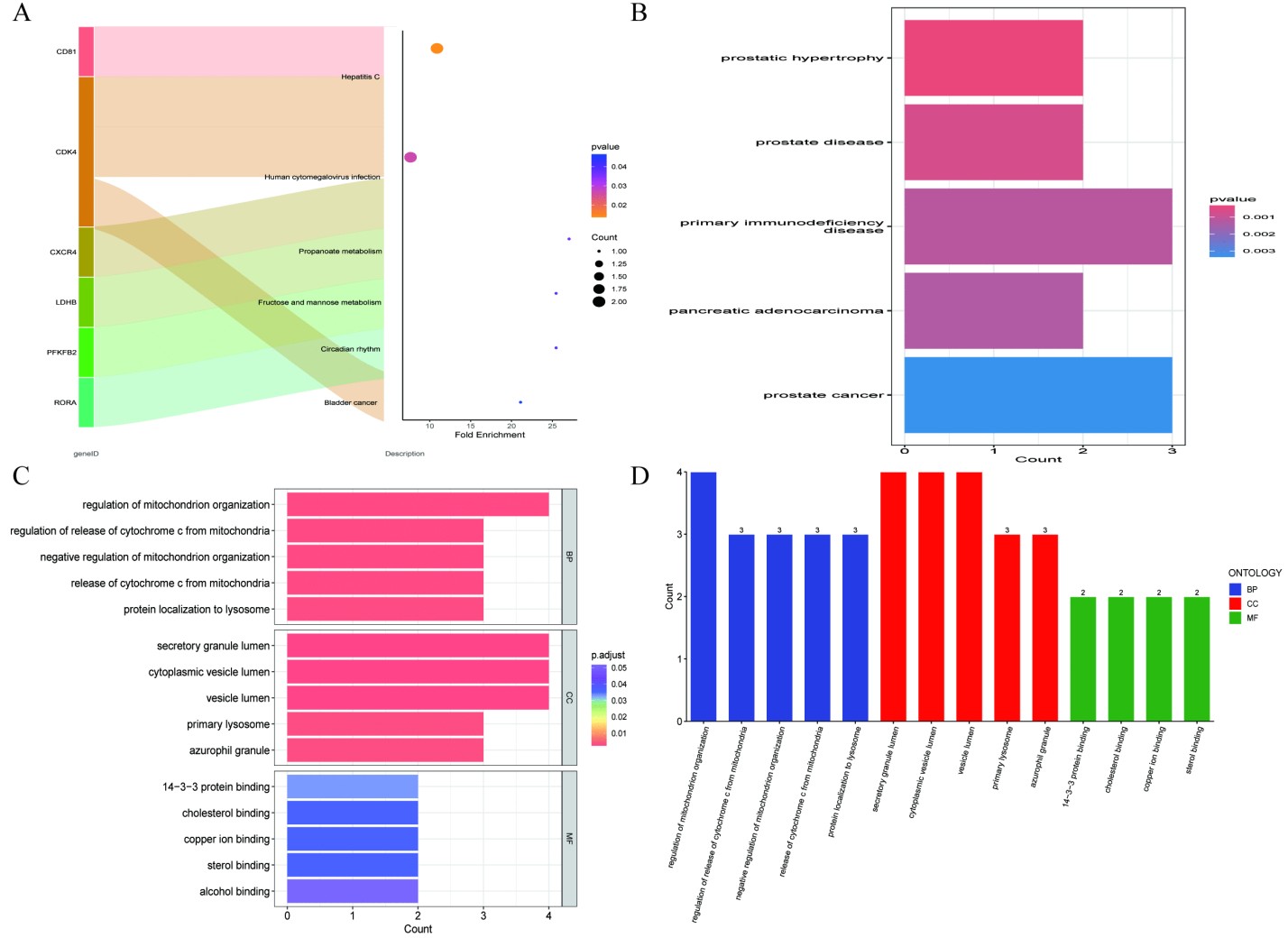

**Fig 2. Functional enrichment analysis of differentially expressed genes. (A)**: Sankey diagram illustrating the KEGG pathway analysis of intersecting genes, highlighting the primary biological processes associated with the shared gene set. **(B)**: Bar chart depicting the results of disease ontology analysis for the intersecting genes, showcasing the significant diseases related to the identified gene set. **(C),(D)**: Bar chart presenting the results of Gene Ontology analysis for the intersecting genes, detailing the biological processes, molecular functions, and cellular components associated with the gene set.

used support vector machines (SVM) to rank features, recursively eliminating lower-weight features to obtain the optimal subset of features. As the number of features varied, the model's generalization error and cross-validation accuracy exhibited fluctuating trends. In the sepsis dataset, the lowest generalization error (0.138) and highest cross-validation accuracy (0.862) were observed when the number of features was twelve (Fig 3I, 3J); similarly, in the atrial fibrillation dataset, the lowest generalization error (0.138) and highest cross-validation accuracy (0.862) were noted when the number of features was nine (Fig 3K, 3L). Ultimately, by taking the intersection of the results from the three machine learning methods, we identified six feature genes in sepsis: CD81, CLU, GLRX, CKAP4, DPEP2, and BCL11B (Fig 3M); and seven feature genes in atrial fibrillation: LDHB, CD81, PFKFB2, CKAP4, CXCR4, DPEP2, and RORA (Fig 3N). After intersecting these two sets of feature genes, we determined CD81, CKAP4, and DPEP2 as key feature genes (Fig 3O)(S1 Table).

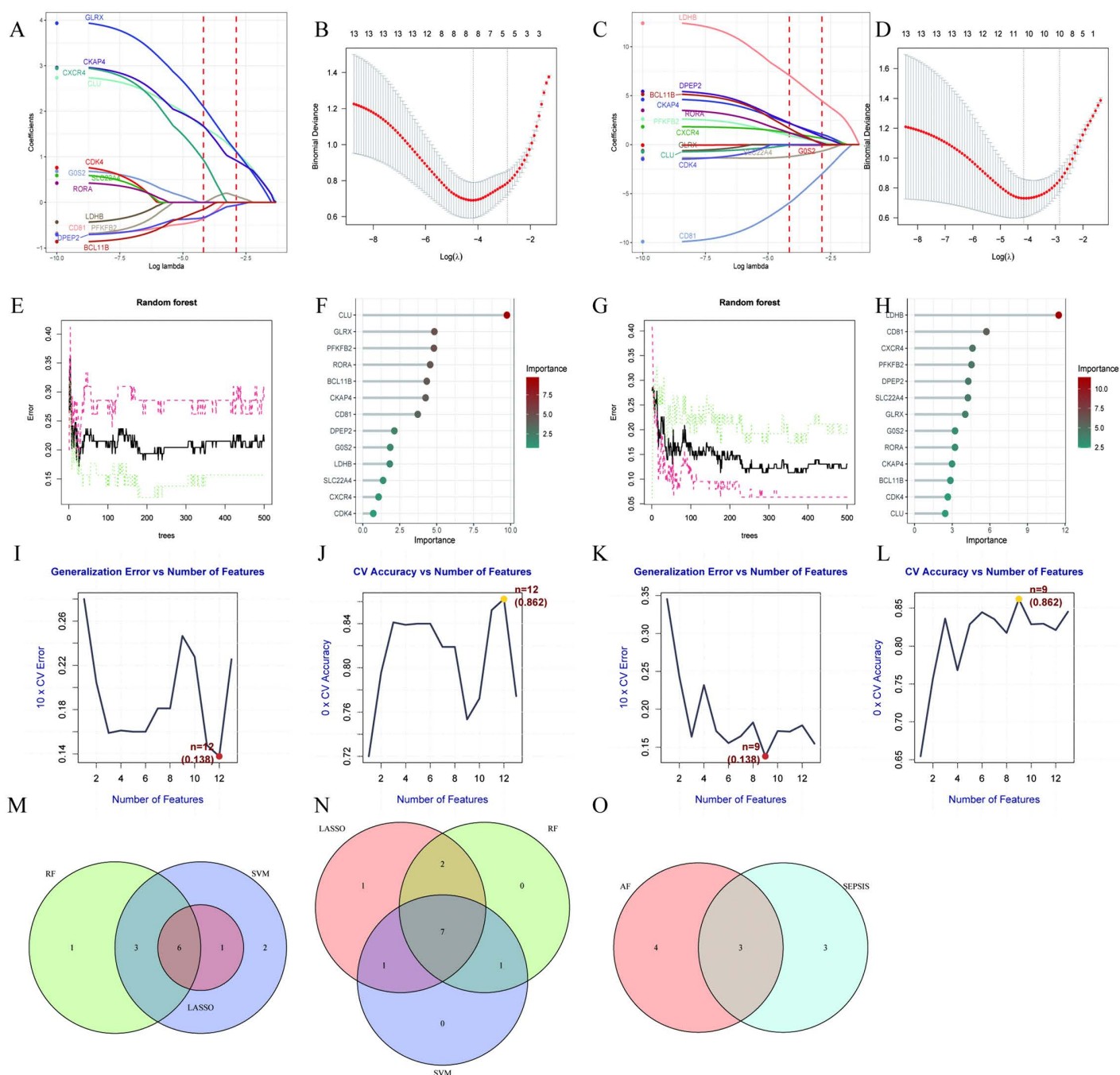

**Fig 3. Integration of three machine learning algorithms to characterize lipid metabolism-associated hub genes in sepsis and atrial fibrillation. (A)**: Lasso regression coefficient path plot for GSE28750 dataset, showing the relationship between Log Lambda and variable coefficients. **(B)**: Cross-validation deviance plot for GSE28750 Lasso regression, depicting Log(λ) values and binomial deviance. **(C)**: Lasso regression coefficient path plot for atrial fibrillation dataset. **(D)**: Cross-validation deviance plot for atrial fibrillation Lasso regression. **(E)**: Random forest error plot for GSE28750 dataset, illustrating tree number versus error rate. **(F)**: Gene importance ranking plot from GSE28750 random forest analysis. **(G)**: Random forest error plot for atrial fibrillation dataset. **(H)**: Gene importance ranking plot from atrial fibrillation random forest analysis. **(I)**: Generalization error plot from SVM-RFE analysis of GSE28750 dataset. **(J)**: Cross-validation accuracy plot from GSE28750 SVM-RFE analysis. **(K)**: Generalization error plot from SVM-RFE analysis of atrial fibrillation dataset. **(L)**: Cross-validation accuracy plot from atrial fibrillation SVM-RFE analysis. **(M)**: Venn diagram of gene intersections from three machine learning methods in GSE28750 dataset. **(N)**: Venn diagram of gene intersections from three machine learning methods in atrial fibrillation dataset. **(O)**: Venn diagram of machine learning identified gene intersections between GSE28750 and atrial fibrillation datasets.

## Validation and evaluation of feature genes

In the sepsis dataset, we constructed receiver operating characteristic (ROC) curves for the feature genes. The results showed that the area under the curve (AUC) for all candidate genes was greater than 0.9, indicating strong diagnostic efficacy (Fig 4A). By further evaluating the expression patterns of the genes across different groups, we observed significant differential expression characteristics (P < 0.0001): CD81, DPEP2, and BCL11B genes were downregulated in the disease group, while CLU, GLRX, and CKAP4 genes were significantly upregulated in the disease group (Fig 4B). In the atrial fibrillation dataset, LDHB, PFKFB2, CKAP4, CXCR4, DPEP2, and RORA genes showed an upregulation trend, while only the CD81 gene was downregulated in the disease group (Fig 4C). The AUC values for all feature genes exceeded 0.6, further validating their potential diagnostic value (Fig 4D). Notably, we found that CD81 was downregulated in both diseases, CKAP4 was consistently upregulated in both diseases, while DPEP2 exhibited a disease-specific expression pattern, downregulated in sepsis and upregulated in atrial fibrillation.To further validate the external generalizability of the model, we constructed predictive models in the external sepsis dataset GSE65682 and the atrial fibrillation dataset. The model AUC for the sepsis dataset was 0.957 (95% confidence interval: 0.916–0.988) (Fig 4E), with individual feature genes also showing AUC values exceeding 0.6 (Fig 4F). The atrial fibrillation dataset model achieved a perfect AUC value of 1.000 (95% confidence interval: 1.000–1.000)(Fig 4G), with individual feature genes also exceeding 0.6 in AUC values (Fig 4H). To assess the predictive performance of the model, we plotted calibration curves to demonstrate the consistency between predicted values and actual observed values. The results indicated a high degree of concordance between our predictive model and the ideal state model (Fig 4I, 4J). Decision curve analysis further revealed the clinical decision-making value of the model, with the distance between the red line and the gray line visually reflecting the accuracy of the model (Fig 4K, 4L). Finally, we constructed scoring scales for these eight feature genes, where the length of the segments represents the contribution of each gene to the model (Fig 4M, 4N).

## Immune cell infiltration analysis

In the immune cell infiltration analysis of the sepsis dataset, we revealed significant changes in the composition of immune cells through heatmaps. In the control group, Neutrophils, Monocytes, and CD4 naive T cells predominated; whereas in the disease group, Neutrophils and Monocytes were more prominent (Fig 5A). There were significant differences in immune cell subpopulations between the two groups, particularly in the following aspects: first, the differences among B cells memory, Macrophages M0, and CD4 naive T cells were extremely significant (P < 0.001). Second, the differences in B cells memory, Plasma cells, NK cells resting, and Neutrophils were also statistically significant (P < 0.01). Finally, the differences in T cells regulatory, Monocytes, Macrophages M1, and activated Dendritic cells also had statistical significance (P < 0.05)(Fig 5B). In the atrial fibrillation dataset, the control group was primarily enriched in Macrophages M2, CD8 T cells, and follicular helper T cells; the disease group was dominated by Macrophages M2, CD8 T cells, and resting Mast cells (Fig 5C). The most significant difference between the two groups was observed in the changes of Macrophages M1 (P < 0.01), followed by differences in resting Mast cells and activated Mast cells (P < 0.05) (Fig 5D). Further analysis of the correlation between sepsis feature genes and immune cells revealed that CLU, GLPX, and CKAP4 were negatively correlated with CD4 naive T cells, while DPEP2 and BCL11B were positively correlated. In Macrophages M0, CD81, DPEP2, and BCL11B were all negatively correlated, with only CKAP4 showing a positive correlation (Fig 5E). In the immune cell correlation analysis of atrial fibrillation feature genes, we observed that CXCR4 was negatively correlated with regulatory T cells and Macrophages M2, while LDHB was positively correlated with gamma delta T cells, and DPEP2 was positively correlated with CD8 T cells (Fig 5F). These findings reveal the complex changes in immune cell infiltration patterns in sepsis and atrial fibrillation, providing new perspectives for understanding the immunological mechanisms of these two diseases. The dynamic changes in different immune cell subpopulations and the differential expression of feature genes not only reflect the immunological characteristics of the diseases but may also provide new potential targets for diagnosis and treatment.

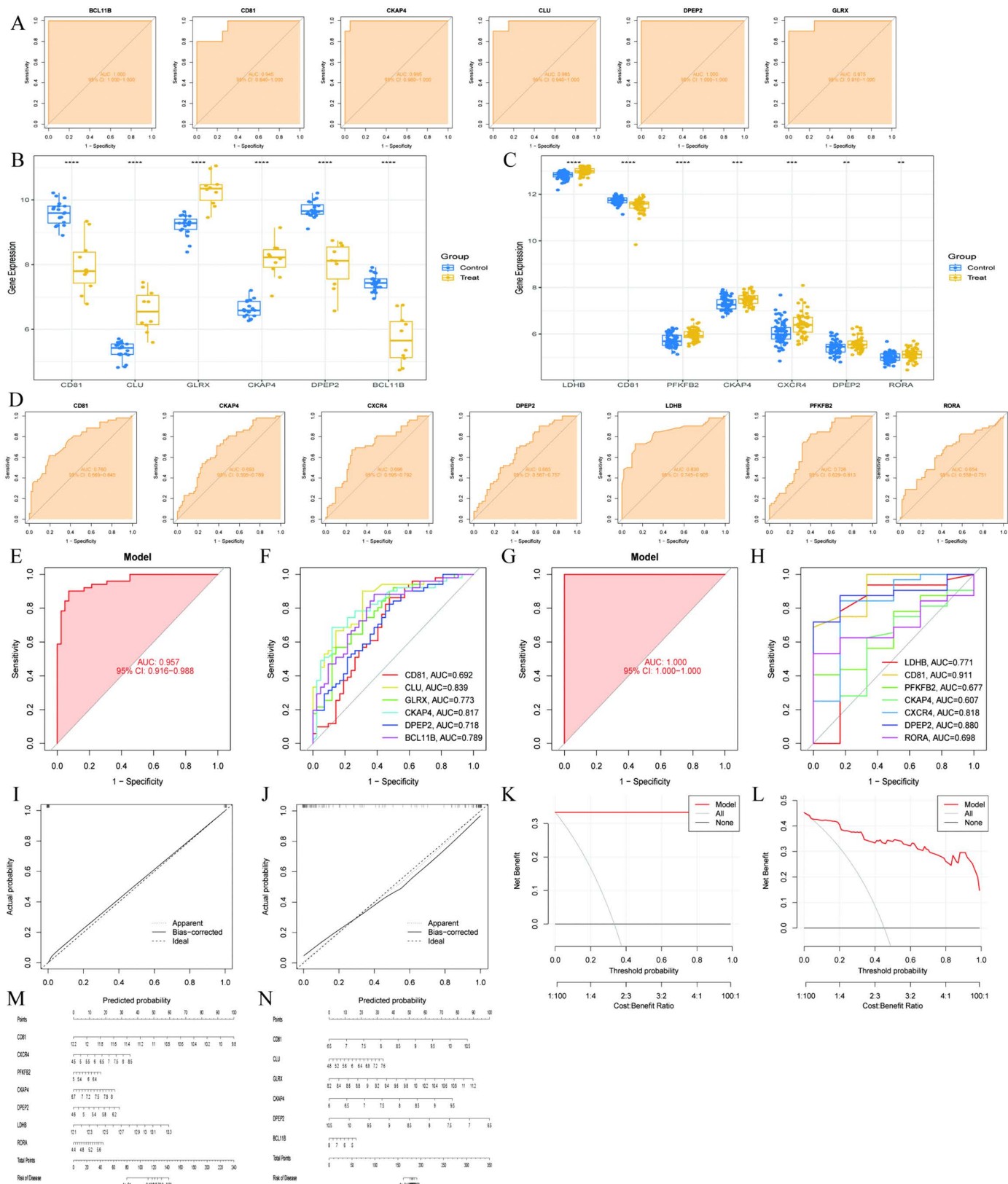

**Fig 4. Experimental and clinical validation of hub genes governing sepsis-atrial fibrillation metabolic crosstalk. (A)**: ROC curve for feature genes in the GSE28750 dataset, showing diagnostic performance. **(B)**: Expression difference plot for feature genes in the GSE28750 dataset. **(C)**:

Expression difference plot for feature genes in the atrial fibrillation dataset. **(D)**: ROC curve for feature genes in the atrial fibrillation dataset. **(E)**: ROC curve from feature genes in the external sepsis dataset GSE65682. **(F)**: Individual feature gene ROC curves in the GSE65682 sepsis dataset. **(G)**: ROC curve from feature genes in the external atrial fibrillation dataset GSE41177. **(H)**: Individual feature gene ROC curves in the GSE41177 atrial fibrillation dataset. **(I,J)**: Calibration curves for sepsis and atrial fibrillation datasets. **(K,L)**: Decision curves for sepsis and atrial fibrillation datasets. **(M,N)**: Nomograms for sepsis and atrial fibrillation datasets showing disease probability.

## GSEA pathway enrichment analysis of feature genes and ceRNA network prediction

In the GSEA pathway analysis of sepsis feature genes, all feature genes were associated with pathways such as Primary immunodeficiency disease, Type 1 diabetes mellitus, Graft-versus-host disease, and antigen processing and presentation (Fig 6A); the feature genes for atrial fibrillation were mainly associated with pathways such as Oxidative phosphorylation, Parkinson disease, and ECM-receptor interaction (Fig 6B). The intersecting pathways of feature genes from both diseases included Antigen processing and presentation, Complement and coagulation cascades, Viral myocarditis, Allograft rejection, and Cell adhesion molecules pathways. These results suggest that the two diseases may share certain immune and pathological mechanisms, providing important clues for further exploration of their molecular basis and potential therapeutic strategies. The competitive endogenous RNA (ceRNA) mechanism refers to a regulatory mechanism in which different types of RNA molecules (such as mRNA, lncRNA, circRNA, etc.) mutually regulate gene expression by competitively binding to shared microRNA (miRNA) binding sites. We analyzed the ceRNA regulatory network of BCL11B, CD81, CKAP4, CLU, GLRX, LDHB, PFKFB2, and RORA, and input the results into Cytoscape software (version 3.10.2) to visually represent the network (Fig 6 C).

## Single-cell transcriptome analysis

We filtered single-cell data from the sepsis dataset GSE167363 and displayed nFeature, nCount, and mitochondrial gene proportions for each sample using violin plots (Fig 7A). Based on the criteria of logFC > 1 and adjusted P value < 0.05, we selected 1,500 highly variable genes while retaining another 18,759 genes (Fig 7B). Next, we performed principal component analysis (PCA) to visualize the position of each cell (Fig 7C). The color of the points in the scatter plot indicates the sample from which the cell originated, with closely clustered points representing higher correlation and proximity indicating stronger association. We plotted scatter plots for the top 25 genes, where the x-axis represents gene scoring values, with larger absolute values indicating stronger correlation (Fig 7D). Additionally, we generated heatmaps displaying the names of the 15 genes with the highest expression levels in each principal component, with deeper colors indicating higher expression intensity (Fig 7E). Subsequently, we used UMAP dimensionality reduction and clustering analysis to categorize cells into 25 subgroups (Fig 8A). Based on gene expression levels within each subgroup, we automatically annotated seven cell types, including B cells, GMP, Monocytes, Neutrophils, NK cells, Platelets, and T cells (Fig 8B). Bar charts illustrated the proportions of different cell types in each sample (Fig 8C). We then input the previously selected sepsis feature genes into the single-cell analysis to assess their expression in various cells. The results showed that BCL11B had the highest expression in T cells, CD81 in natural killer cells, CKAP4 in Neutrophils, CLU in Platelets, while DPEP2 and GLRX exhibited the highest expression levels in Monocytes (Fig 8D). Finally, we combined the data from the previous immune infiltration analysis to plot the correlation between feature genes and immune cells. The analysis indicated that BCL11B, CD81, CKAP4, and GLRX were positively correlated with immune cells, while CLU and DPEP2 primarily exhibited negative correlations (Fig 8E). These results provide an important molecular biological basis for our understanding of the functions of immune cells and the roles of feature genes in sepsis.

## Exploratory compound enrichment analysis

To explore potential compound-gene associations, we input the key feature genes of sepsis and atrial fibrillation into the DSigDB database via the Enrichr online platform (https://maayanlab.cloud/Enrichr/) for gene-compound enrichment

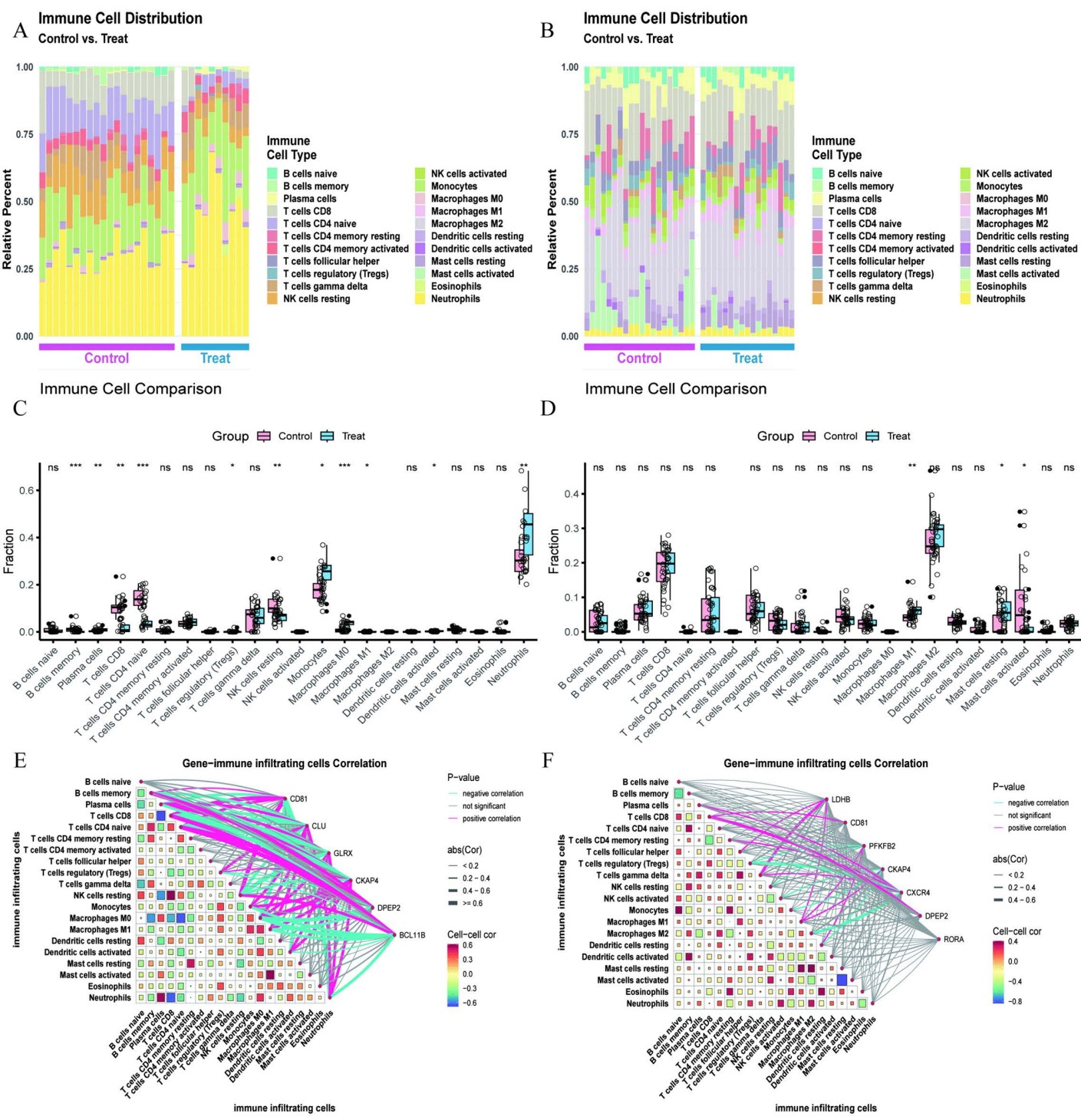

**Fig 5. Comprehensive immune profiling reveals infiltration dynamics and hub gene-immune cell interactions in sepsis and atrial fibrillation cohorts. (A,B)**: Immune cell infiltration heatmaps comparing control and disease groups in sepsis and atrial fibrillation datasets, illustrating the distribution and composition of immune cells. **(C,D)**: Differential analysis of immune cell populations in control and disease groups for sepsis and atrial fibrillation datasets. **(E,F)**: Correlation analysis between feature genes and different immune cell types in sepsis and atrial fibrillation datasets.

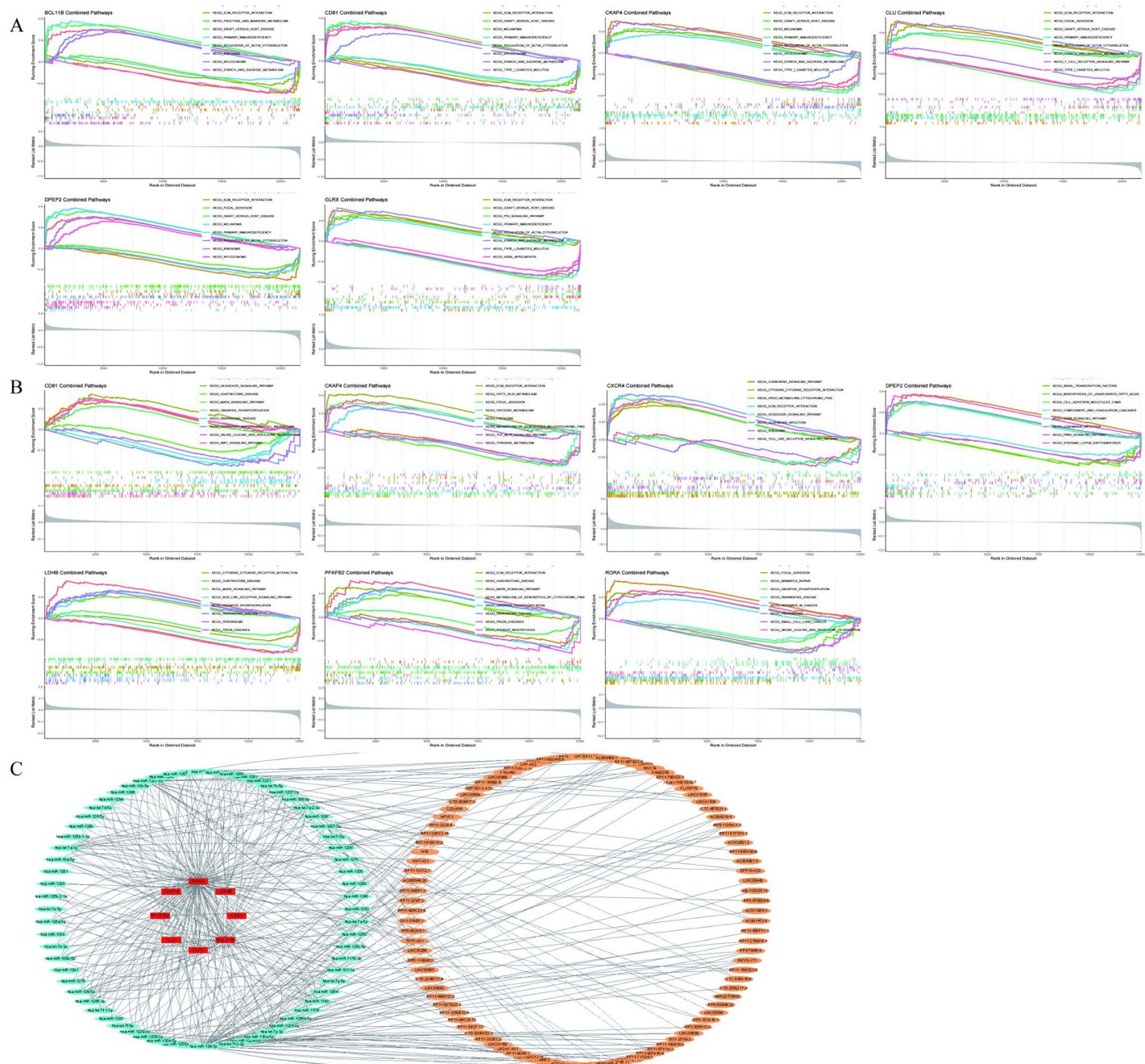

**Fig 6. Integrated GSEA pathway annotation and ceRNA network construction for mechanistic characterization of hub genes. (A)**: Gene Set Enrichment Analysis (GSEA) pathway enrichment plot for upregulated and downregulated pathways in sepsis-related feature genes. **(B)**: GSEA pathway enrichment plot for upregulated and downregulated pathways in atrial fibrillation-related feature genes. **(C)**: Competitive endogenous RNA (ceRNA) network diagram for feature genes from both diseases.

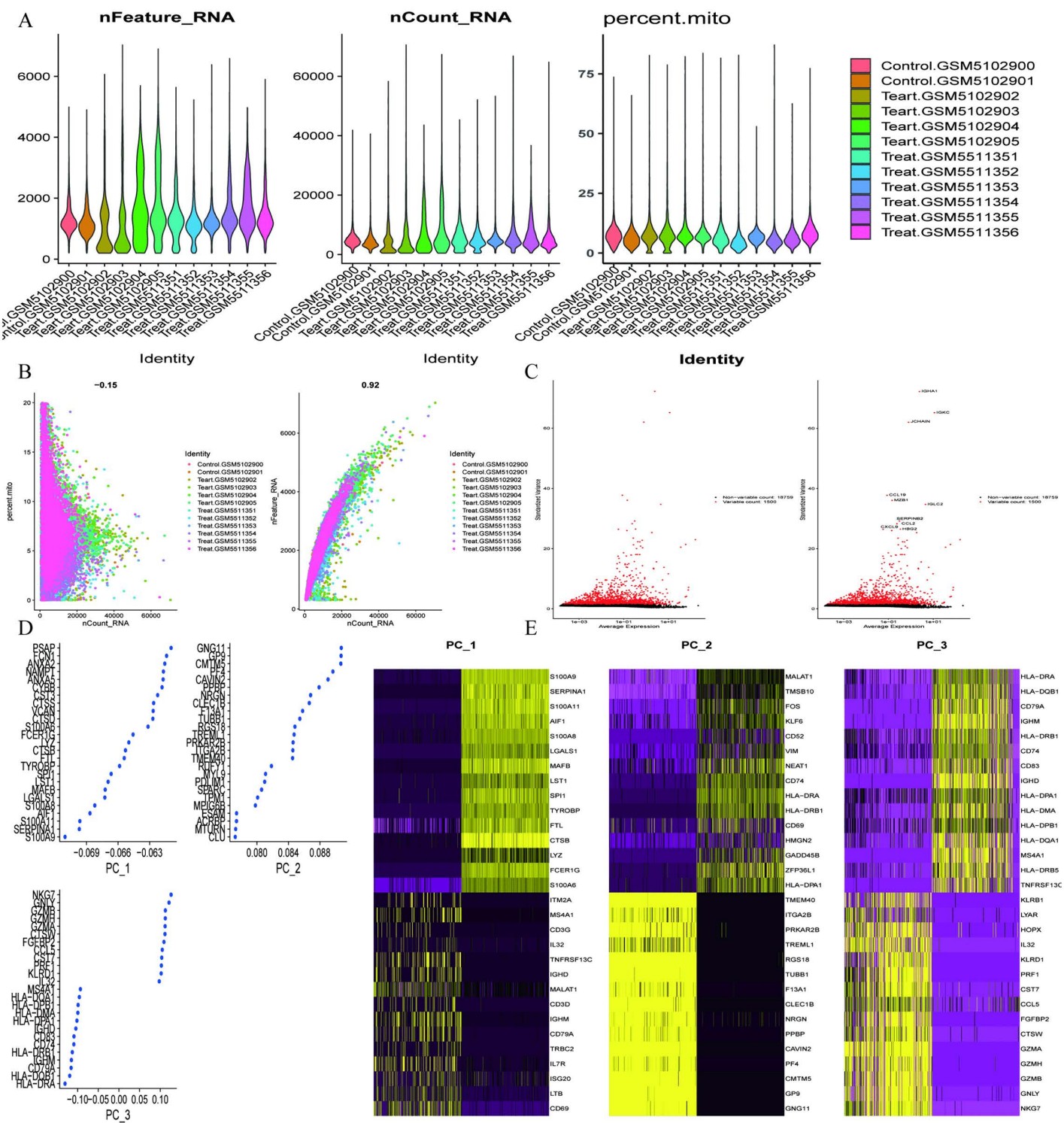

**Fig 7. Comprehensive transcriptomic landscape of sepsis dataset GSE167363: Integrated quality metrics, PCA decomposition, and differential gene expression dynamics. (A)**: Violin plot of RNA features, RNA counts, and mitochondrial gene proportion across samples in the sepsis dataset GSE167363. **(B)**: PCA distribution plot for samples in the sepsis dataset GSE167363. **(C)**: Differential gene expression scatter plot for the sepsis dataset GSE167363. **(D)**: Scatter plot of top 25 differential genes for each PCA component in the sepsis dataset GSE167363. **(E)**: Heatmap of top 15 differential genes for each PCA component in the sepsis dataset GSE167363.

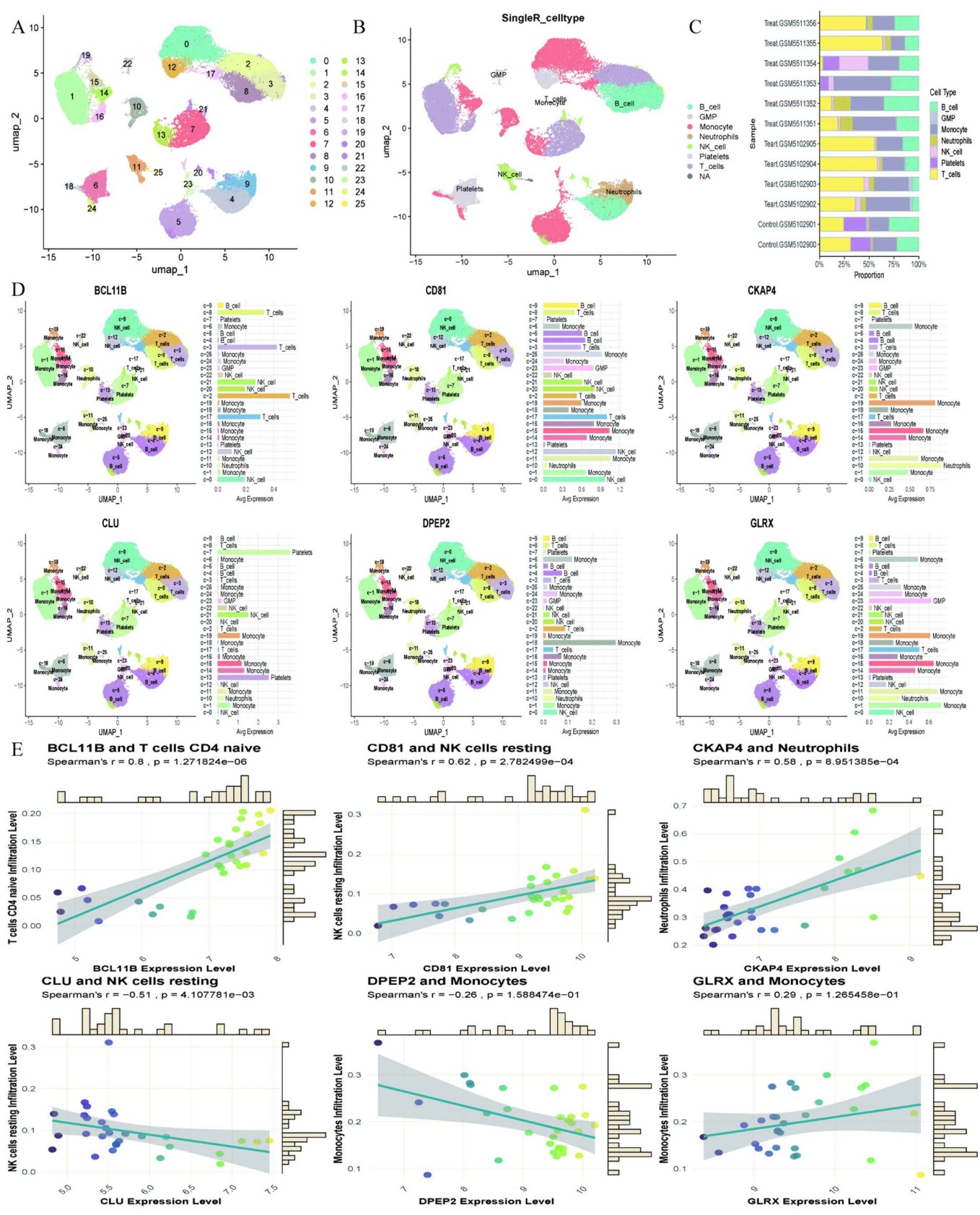

**Fig 8. Single-cell dissection of sepsis immune microenvironment: Cellular heterogeneity, feature gene spatial dynamics, and immune-centric molecular crosstalk. (A)**: UMAP dimensionality reduction plot from single-cell analysis of the sepsis dataset. **(B)**: Automated cell type annotation plot from single-cell analysis. **(C)**: Distribution of sepsis feature genes in UMAP dimensionality reduction plot. **(D)**: Correlation plot between sepsis feature genes and different immune cells.

analysis. This exploratory analysis yielded ten candidate compounds with statistical associations (P < 0.05), including methotrexate, deptropine, 2-Naphthoxyacetic acid, coenzyme A, etretinate, methyprylon, retinol, nitroglycerin, GLYCO-PROTEIN, and diazepam (Table 2). These hits are presented as preliminary hypotheses and require further validation.

## Discussion

Lipid metabolism is a critical physiological process for maintaining homeostasis, and its dysregulation not only reflects pathological changes in diseases but may also serve as a key driving factor in disease onset and progression. Sepsis and AF, as complex diseases that severely threaten human health, have long been considered to have independent patho-physiological mechanisms. However, increasing evidence suggests that abnormalities in lipid metabolism may represent an important molecular bridge connecting these two diseases. This study systematically reveals the shared molecular basis of lipid metabolism in sepsis and AF by identifying 13 differentially expressed genes, providing new molecular insights into the potential associations between these two diseases.Biological pathway and functional enrichment analyses are essential tools for elucidating the molecular mechanisms of diseases. In complex inflammatory diseases like sepsis and AF, abnormalities in a single pathway often fail to fully explain the mechanisms of disease onset. This study aims to reveal the complex regulatory networks of lipid metabolism in these two diseases from a systems biology per-spective by analyzing the enrichment of differentially expressed genes across various functions and pathways. During the pathophysiological processes of sepsis and AF, the release of cytochrome c from mitochondria is a key step in regulating apoptosis and mitochondrial dysfunction [21]. In sepsis, mitochondrial dysfunction and excessive reactive oxygen species (ROS) production lead to the release of cytochrome c, promoting the activation of apoptotic signaling pathways, which in turn causes myocardial cell dysfunction and multiple organ failure [22]. Similarly, the release of cytochrome c is also criti-cal in AF, often associated with myocardial cell apoptosis and mitochondrial dysfunction. This release is typically triggered by oxidative stress, inflammatory responses, and abnormal activation of intracellular signaling, leading to the activation of apoptotic proteins such as caspase-3, thereby promoting myocardial cell apoptosis and atrial remodeling [23]. In sepsis, fructose disrupts the intestinal barrier, inducing endotoxemia and exacerbating liver inflammation and lipid metabolism disorders [24]. In AF, fructose alters calcium homeostasis and induces oxidative stress, leading to structural remodeling of atrial myocytes, with its metabolic product F1P potentially promoting atrial fibrosis via the TGF-β signaling pathway

**Table 2. Exploratory compound hits from DSigDB enrichment analysis.**

| Term | P-value | Combined Score | Genes |
|------|---------|----------------|-------|
| methotrexate | 0.0028101450518222197 | 369.0378659716684 | CD81;CKAP4 |
| deptropine | 0.005091520900981562 | 1597.1740514070286 | CKAP4 |
| 2-Naphthoxyacetic acid | 0.005689387732894645 | 1394.274144706199 | CKAP4 |
| coenzyme A | 0.008824262742308117 | 813.0729238663458 | CKAP4 |
| etretinate | 0.013291221555431713 | 488.7491512268038 | CKAP4 |
| methyprylon | 0.01893008279203636 | 312.81077549345065 | CKAP4 |
| retinol | 0.020114467992547895 | 289.5192282207325 | CKAP4 |
| nitroglycerin | 0.024104731206729083 | 229.49063947550405 | CKAP4 |
| GLYCOPROTEIN | 0.028084148529689098 | 188.21447440701544 | CD81 |
| diazepam | 0.02867277029258455 | 183.18631536211723 | CKAP4 |

[25]. Furthermore, fructose-induced dysbiosis exacerbates systemic inflammatory responses [26]. Mannose-binding lectin (MBL) plays a crucial role in both sepsis and AF. In sepsis, MBL gene polymorphisms affect susceptibility to infections and prognosis, and its complement activation and inflammatory regulatory functions are vital for combating infections [27]. In AF, mannose metabolites such as mannose-6-phosphate are involved in cardiac structural regulation, and MBL deficiency may increase the risk of cardiovascular remodeling [28]. Sepsis is characterized by a systemic inflammatory response, and studies indicate that HDL not only plays a key role in cholesterol transport but also neutralizes the inflammatory effects of pathogen-associated lipids (such as lipopolysaccharides, LPS) by binding to them, thereby alleviating disease progression [29]. HDL promotes the clearance of LPS, inhibits immune stimulation, and reduces the release of pro-inflammatory cytokines, improving prognosis [30]. In the context of AF, HDL's cholesterol efflux capacity (CEC) is closely related to structural remodeling of the left atrium. HDL dysfunction, particularly weakened CEC, is associated with left atrial enlargement and fibrosis, with inflammation and oxidative stress considered major factors leading to this dysfunction [31]. This decline in HDL function may trigger structural changes in the atrium, increasing the risk of AF [32]. In the plasma of sepsis patients, particularly neutrophil-derived extracellular vesicles (such as elongated neutrophil-derived structures, ENDS) are significantly increased, with numbers 10–100 times higher than those in healthy individuals. The concentration of the S100A8–S100A9 complex, which is related to inflammation and coagulation, is positively correlated with these vesicles, suggesting that they promote inflammation and coagulation in the pathophysiology of sepsis [33]. In AF, the levels of extracellular vesicles carrying tissue factor (TF) are significantly elevated, which is associated with hemodynamic changes and endothelial dysfunction caused by AF. Blood stasis induced by AF promotes the release of TF-rich vesicles, increasing the overall tendency for coagulation [34].

In the era of big data and artificial intelligence, machine learning has become an indispensable tool in biomedical research. Traditional statistical methods often struggle to handle high-dimensional and complex biological data. This study employs advanced machine learning algorithms such as Lasso regression, random forests, and support vector machines to accurately identify key feature genes associated with sepsis and AF from differentially expressed genes. CD81 is a tetraspanin protein that is widely involved in cell signaling and lipid metabolism. In lipid metabolism, CD81 promotes the uptake and metabolism of fatty acids by interacting with membrane proteins such as CD36, particularly in brown adipocytes, where its signaling function is closely related to cell proliferation and lipid release [35]. In sepsis, CD81 expression is significantly increased, and its role in exosomes suggests a close association with inflammatory responses and immune regulation. CD81 may regulate immune cell activation and intercellular signaling by influencing exosomal components and release [36]. Regarding AF, CD81 expression in cardiac tissue is associated with electrophysiological properties and inflammatory states. It may affect cardiac remodeling and electrophysiological stability by regulating cardiac cell signaling and intercellular interactions [37]. CKAP4 (cytoskeleton-associated protein 4) has garnered increasing attention for its role in lipid metabolism, sepsis, and AF. Research indicates that CKAP4 may play a crucial role in lipid metabolism by regulating the transport and oxidation of fatty acids. It is involved in the dynamic balance of intracellular lipids, affecting the differentiation and function of adipocytes, thus playing a key role in energy metabolism and lipid storage [38]. Furthermore, CKAP4 expression is closely related to inflammatory responses, making it an important player in the pathophysiology of sepsis. As a systemic inflammatory response, sepsis may influence its severity and prognosis by regulating immune cell function and the release of inflammatory mediators [39]. In AF, CKAP4's role may be related to its effects on cardiac electrophysiological properties and myocardial cells [40]. Studies have shown that CKAP4 may influence cardiac electrical activity by regulating calcium channels and the stability of the cytoskeleton in myocardial cells, thereby playing a role in the occurrence and maintenance of AF [41]. Dipeptidase-2 (DPEP2) is increasingly recognized for its role in lipid metabolism, sepsis, and AF, primarily functioning through the regulation of inflammation and cell signaling. As a membrane-bound dipeptidase, DPEP2 plays a key role in leukotriene metabolism, catalyzing the conversion of leukotriene D4 to leukotriene E4, thereby modulating immune responses [42]. In sepsis, DPEP2 is closely related to macrophage function, and its deficiency leads to excessive production of pro-inflammatory factors (such as TNF-α and IL-6), highlighting its importance

in inhibiting macrophage overactivation and maintaining immune homeostasis [43]. DPEP2 may limit inflammatory responses in sepsis by inhibiting the NF-κB and p38 MAPK signaling pathways. Its regulation of lipid metabolism may influence cardiac electrophysiological properties, and its upregulation in cardiac macrophages may be associated with the regulation of cardiac inflammatory responses [44]. The complement system, as a component of innate immunity, plays a crucial role in pathogen clearance and inflammatory responses. In metabolic disorders such as gestational diabetes mellitus (GDM), changes in complement components like C3 and C4 levels are closely related to lipid metabolism abnormalities, promoting low-grade chronic inflammation, vascular injury, and thrombosis, thereby increasing the risk of cardiovascular diseases [45]. In sepsis, dysregulation of the complement and coagulation systems is a key factor in multiple organ failure. Systemic inflammation leads to excessive activation of the complement and coagulation cascades, resulting in disseminated intravascular coagulation (DIC) and tissue damage, thereby increasing mortality risk [46]. Therefore, interventions such as complement inhibitors may have potential clinical significance in the treatment of sepsis. In AF, electrophysiological remodeling of the left atrium is associated with dysregulation of coagulation and the complement system. AF patients often exhibit a hypercoagulable state, increasing the risk of thrombosis and ischemic stroke. Studies have found that levels of complement components C1q and factor H are reduced in AF patients, which is associated with impaired immune responses and tissue damage, suggesting that the interaction between the coagulation and complement systems plays an important role in the pathology of AF [47].

The analysis of immune cell infiltration reveals significant differences in immune cell composition between patients with sepsis and AF, particularly in the proportions of neutrophils and monocytes. These changes are closely related to abnormalities in lipid metabolism, suggesting that lipid metabolism not only affects cardiac structure but may also influence disease progression by regulating immune responses. Neutrophils play a crucial role in lipid metabolism, sepsis, and AF, particularly in inflammation regulation and immune function. As the body's first line of defense against infection, neutrophils capture and kill pathogens by releasing neutrophil extracellular traps (NETs), but excessive NET formation is associated with various pathological states, including sepsis and AF [48]. In sepsis, dysfunctional neutrophils exacerbate inflammatory responses. Studies have shown that neutrophil activation and NET release are significantly increased in sepsis patients, leading to tissue damage and organ dysfunction. NETs promote inflammation and coagulation responses in endothelial cells, further exacerbating the pathological process and being closely related to lipid metabolism, affecting the activation state of neutrophils [49]. In AF, neutrophils and their released NETs are considered potential mechanisms for atrial remodeling and electrophysiological instability. NETs can induce mitochondrial damage and autophagic cell death in myocardial cells, increasing the risk of AF [50]. The concentration of NETs is significantly elevated in the peripheral blood and atrial tissue of AF patients, indicating their potential key role in pathophysiology [51]. Monocytes play an important regulatory role in the pathophysiology of lipid metabolism, sepsis, and AF. They participate in the regulation of inflammatory responses by producing pro-inflammatory and anti-inflammatory factors. In lipid metabolism, the ratio of monocytes to HDL-C is an important marker reflecting inflammation and oxidative stress. In AF, lower HDL-C levels are associated with an increased risk of AF, and the accumulation of monocytes may promote atrial remodeling by releasing activated substances, thereby influencing the occurrence of AF [52]. In sepsis, monocytes exhibit spontaneous activation and superoxide release but show low reactivity when stimulated, forming a state of "tolerance," reflecting the body's regulation of excessive inflammation [53]. Additionally, monocytes are associated with increased left atrial diameter (LAD), suggesting that they may participate in the occurrence of AF by promoting atrial fibrosis. Therefore, the combination of MHR and LAD could serve as predictive indicators for recurrence after radiofrequency ablation in AF patients [54].

Lipid metabolism abnormalities have been confirmed to play a key regulatory role in the pathogenesis of various diseases, yet the molecular crosstalk between small molecules and these lipid metabolism-driven pathological processes remains incompletely elucidated. The exploratory compound enrichment analysis in this study, anchored in key feature genes, is not intended to identify therapeutic agents but rather to generate preliminary hypotheses for investigating potential small molecule-gene interaction networks. Coenzyme A (CoA)—one of the hits from the DSigDB

enrichment—functions as a core cofactor in lipid metabolic cascades, serving as an essential mediator of fatty acid synthesis and β-oxidation; its dysregulation directly disrupts metabolic homeostasis. For instance, in subclinical ketosis, downregulation of acetyl-CoA acetyltransferase 2 (ACAT2) (a CoA-dependent enzyme) leads to hepatic triglyceride accumulation and impaired cholesterol synthesis, underscoring how CoA-related molecular events drive metabolic disorders [55]. In sepsis, CoA participates in the regulation of hepatic metabolism by modulating the expression of bile acid synthesis-related enzymes such as bile acid-CoA-amino acid N-acyltransferase (rBAT), with dysregulation potentially contributing to liver dysfunction and cholestasis—findings that reflect CoA's complex role in coordinating inflammatory and metabolic stress responses [56]. In the context of AF, CoA metabolic pathways have also been implicated in pathological processes, as evidenced by previous studies noting that statins modulate CoA-related cascades. This existing evidence provides a reference framework for exploring whether the compounds enriched in our analysis might interact with feature genes via analogous metabolic or signaling pathways [57]. Importantly, these compound hits are derived from in silico enrichment alone, lack experimental validation of interaction specificity. Their value lies solely in guiding subsequent mechanistic investigations of small molecule-gene-lipid metabolism regulatory networks.

This study systematically explores the relationship between sepsis and AF, emphasizing the importance of lipid metabolism, identifying 13 differentially expressed genes, and providing new perspectives. By utilizing various bioinformatics tools and machine learning algorithms, three key genes were selected, ensuring the reliability of the results and revealing the shared role of lipid metabolism abnormalities in these two diseases while pointing to new therapeutic targets. Additionally, the study identified candidate drugs with therapeutic potential, providing new directions for intervention. It should be noted that this study has certain limitations: First, there are variations in cohort sizes—for instance, the sepsis validation set (GSE65682) has a relatively large sample size (750 cases), while the AF validation set (GSE41177) has a smaller sample size (14 cases). Moreover, heterogeneity exists across detection platforms (e.g., GPL570, GPL96) and tissue sources (whole blood, myocardial tissue) in some datasets, which may impact the generalizability of the results. Second, wet experimental validation using clinical samples (e.g., qPCR, immunohistochemistry) is lacking, and the exploration of the biological functions of feature genes and lipid metabolism-related mechanisms needs further refinement. Third, the single-cell transcriptome analysis (based on GSE167363) involves a small sample size (10 cases, 2 controls), which may lead to biases in cell typing. Future studies should expand the sample size and conduct multi-center, multi-platform research, while standardizing tissue detection types to reduce the interference of heterogeneity. Simultaneously, experimental validation using clinical samples should be performed to confirm the expression patterns and functions of feature genes. Furthermore, in-depth analysis is required to clarify the molecular mechanisms by which lipid metabolism abnormalities regulate sepsis and AF through specific signaling pathways (e.g., NF-κB, p38 MAPK), evaluate the clinical translational value of feature genes and candidate drugs, and explore personalized treatment strategies based on lipid reprogramming. Ultimately, this will provide more reliable evidence for the early diagnosis, risk stratification, and precision treatment of these two diseases.

## Conclusion

This study systematically reveals the central regulatory role of lipid metabolism dysregulation in the pathogenesis of sepsis and AF by integrating bioinformatics and machine learning methods. We identified 13 differentially expressed genes, including key feature genes such as CD81, CKAP4, and DPEP2. Functional enrichment analysis indicates that these genes are primarily involved in mitochondrial function regulation, complement-coagulation cascades, and cell adhesion molecule pathways, suggesting a shared molecular basis of immune-metabolic dysregulation between the two diseases. In sepsis, abnormalities in lipid metabolism exacerbate endothelial injury and organ dysfunction by promoting neutrophil/monocyte infiltration, exosome release, and inflammatory cytokine storms. In AF, these abnormalities lead to electrical remodeling by affecting myocardial calcium homeostasis, oxidative stress, and atrial fibrosis. Immune infiltration analysis further confirms the critical roles of NETs and the monocyte-to-high-density lipoprotein cholesterol ratio (MHR) in disease

progression. Exploratory compound enrichment analysis identified candidate hits associated with key feature genes, and these findings constitute hypotheses for future in-depth investigations into intermolecular interactions. This research provides a theoretical framework of "lipid metabolism-inflammation-organ injury" for the comorbid mechanisms of these two diseases, but further validation of its translational value through expanded clinical samples and functional experiments is necessary. Future exploration of personalized treatment strategies based on lipid reprogramming is warranted.

## Supporting information

**S1 Fig. Flow-chart of dataset analysis in this paper.**
(TIF)

**S1 Table. The expression profiles of common key genes across the two diseases.**
(DOCX)

## Author contributions

**Conceptualization:** Changze Ou, Huajun Long.

**Data curation:** Changze Ou, Binbin Chen.

**Funding acquisition:** Huajun Long.

**Methodology:** Changze Ou, Haidong Yu.

**Software:** Changze Ou, Haidong Yu.

**Writing – original draft:** Changze Ou, Binbin Chen.

**Writing – review & editing:** Changze Ou, Huajun Long.

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
