## [Decision Letter · Decision Letter 0]

16 Sep 2025

Dear Dr. Long,

Thank you for submitting your manuscript to PLOS ONE. After careful consideration, we feel that it has merit but does not fully meet PLOS ONE’s publication criteria as it currently stands. Therefore, we invite you to submit a revised version of the manuscript that addresses the points raised during the review process.

We look forward to receiving your revised manuscript.

Kind regards,

Wenxing Li

Academic Editor

PLOS ONE

Journal Requirements:

“This study was supported by the National Traditional Chinese Medicine Expert Inheritance Studio Construction Project (Document No. [2022] 75 from the National Administration of Traditional Chinese Medicine), and the Scientific Research Plan Project of the Health Commission of Hunan Province (Grant No. C202303078160).” 

“This study was supported by the National Traditional Chinese Medicine Expert Inheritance Studio Construction Project (Document No. [2022] 75 from the National Administration of Traditional Chinese Medicine), and the Scientific Research Plan Project of the Health Commission of Hunan Province (Grant No. C202303078160).” 

“This study was supported by the National Traditional Chinese Medicine Expert Inheritance Studio Construction Project (Document No. [2022] 75 from the National Administration of Traditional Chinese Medicine), and the Scientific Research Plan Project of the Health Commission of Hunan Province (Grant No. C202303078160).”

“This study was supported by the National Traditional Chinese Medicine Expert Inheritance Studio Construction Project (Document No. [2022] 75 from the National Administration of Traditional Chinese Medicine), and the Scientific Research Plan Project of the Health Commission of Hunan Province (Grant No. C202303078160).”

Reviewers' comments:

Reviewer's Responses to Questions

**Comments to the Author**

1. Is the manuscript technically sound, and do the data support the conclusions?

Reviewer #1: Yes

Reviewer #2: Yes

2. Has the statistical analysis been performed appropriately and rigorously?

Reviewer #1: Yes

Reviewer #2: I Don't Know

3. Have the authors made all data underlying the findings in their manuscript fully available?

Reviewer #1: Yes

Reviewer #2: Yes

4. Is the manuscript presented in an intelligible fashion and written in standard English?

Reviewer #1: Yes

Reviewer #2: Yes

Reviewer #1: This manuscript explores shared molecular links between lipid-metabolism dysregulation, sepsis, and atrial fibrillation (AF) using public transcriptomic datasets, differential expression, WGCNA, multiple ML feature-selection strategies, immune-cell deconvolution, single-cell context, and drug-repurposing enrichment. CD81, CKAP4, and DPEP2 emerge as convergent hub genes with promising discriminative performance in held-out datasets. The cross-disease angle is timely, and the multi-modal analytic approach is a strength.

The work is thoughtfully designed and potentially impactful. Most of my comments are clarification and presentation tweaks that, if addressed, would improve transparency and reader confidence without requiring heavy re-analysis. I particularly appreciate the effort to bring single-cell data and external evaluations into the narrative. Below are suggestions to strengthen the manuscript:

1. Clarify dataset flow and harmonization: add a one-page flow diagram and briefly describe batch correction or use of study covariates. A simple schematic that lists platform, tissue, n, discovery vs validation, and QC would help a lot.

2. Standardize or justify DEG thresholds: align to a common |logFC| (for example, ≥0.5) or add a short rationale plus a quick sensitivity check to show stability of key results.

3. Specify and lock the validation pipeline: say where feature selection sits relative to cross-validation, how hyperparameters were tuned, and confirm that external sets were untouched until final testing. Add PR-AUC and a simple calibration metric (Brier) in the supplement. A brief leave-one-study-out check would be a nice bonus.

4. Validate the lipid gene universe: repeat enrichment and selection with curated Reactome or GO or KEGG lipid sets and report whether top pathways and hubs remain consistent. A short methods note plus a small supplemental table is enough.

5. Detail immune deconvolution and scRNA-seq QC: name the signature or tool, note how multiple testing across cell types was handled, and list basic scRNA thresholds (UMIs or genes, mitochondrial percent, doublet handling). One concise paragraph in Methods will do.

6. Reframe drug-enrichment as exploratory: present DSigDB or Enrichr hits as hypotheses and, if possible, cross-check with an orthogonal resource such as L1000 or CMap. Avoid therapeutic language.

7. Qualify performance claims: keep the strong AUCs but add a brief caveat about cohort size and platform or tissue heterogeneity. Include study-wise curves in the supplement so readers can see consistency.

8. Share reproducible materials: deposit analysis code, processed matrices, and per-figure numeric data in a public repository (GitHub). This will satisfy transparency expectations and help others reuse the work.

The manuscript is well conceived and largely sound. My suggestions are to clarify the dataset flow, lightly harmonize thresholds, document the ML validation steps, and moderate the interpretation. These can be handled with concise text edits and small supplemental additions, not major new analyses. With these clarifications, the work should meet PLOS ONE’s standards. Thank you for the opportunity to review this manuscript.

Reviewer #2: The study presents a well-structured and methodologically sound investigation into lipid metabolism abnormalities in sepsis and atrial fibrillation (AF). The integration of multiple GEO datasets and the use of machine learning models (LASSO, Random Forest, SVM-RFE) are appropriate and executed with rigor. The identification of CD81, CKAP4, and DPEP2 as shared biomarkers is supported by robust statistical validation.

Suggestions:

• Clarify the independence of training and validation datasets to address potential overfitting, especially given the perfect AUC (1.000) reported for AF.

• Consider including experimental validation (e.g., qPCR or immunohistochemistry) to strengthen biological relevance.

2. Methodological Rigor

The manuscript demonstrates strong computational methodology. Differential expression analysis, WGCNA, and enrichment analyses are well-documented. Immune infiltration and single-cell transcriptomic analyses add depth to the findings.

Suggestions:

• Provide more detail on batch effect correction and normalization steps for reproducibility.

• Include rationale for selecting top 4,000 lipid metabolism genes from GeneCards—this cutoff appears arbitrary without justification.

**Do you want your identity to be public for this peer review?** For information about this choice, including consent withdrawal, please see our Privacy Policy

Reviewer #1: No

Reviewer #2: **Yes: ** Maha Ahmed

---

## [Author Response · Author response to Decision Letter 1]

5 Oct 2025

Response to Reviewers' Comments – Manuscript ID [PONE-D-25-28089R1]

Dear Reviewer1,

Thank you for your valuable feedback and constructive suggestions on our manuscript titled "Molecular Mechanisms of Lipid Metabolism Abnormalities Driving Sepsis and Atrial Fibrillation: A Systematic Study Based on Bioinformatics and Machine Learning". We have carefully revised the manuscript in accordance with your recommendations. Below, we provide a detailed point-by-point response to your comments.

1. Clarify dataset flow and harmonization: add a one-page flow diagram and briefly describe batch correction or use of study covariates. A simple schematic that lists platform, tissue, n, discovery vs validation, and QC would help a lot.

Response: To address this issue, we have supplemented Supplementary Figure 1, which intuitively illustrates the complete analytical workflow, ranging from dataset acquisition, QC, and batch correction to model training and validation. Concurrently, we have updated Table 1 (Dataset Information) to explicitly add "Tissue Source" (e.g., whole blood for sepsis datasets, right atrial appendage/left atrial appendage for AF datasets) and "Cohort Role" (discovery/training set, validation set, single-cell analysis set) for each GSE series. For scRNA-seq data (GSE167363), we have refined the QC criteria in the "Single-Cell Transcriptome Analysis" section of the Methods. Specifically, we have clarified the thresholds for cell retention (e.g., nFeature_RNA > 300, mitochondrial gene proportion < 20%) and the rules for outlier exclusion. Regarding batch correction, additional details have been supplemented in the "Batch-Corrected Integration of AF Datasets" section: the removeBatchEffect function from the R package limma was employed, with "dataset origin + detection platform" set as covariates to eliminate systematic biases. The effectiveness of batch correction was verified using PCA and box plots of gene expression distribution.

2. Standardize or justify DEG thresholds: align to a common |logFC| (for example, ≥0.5) or add a short rationale plus a quick sensitivity check to show stability of key results.

Response: The threshold setting for DEGs was based on the unique transcriptomic characteristics of the two diseases, which has been further elaborated in the "Differential Analysis of Gene Expression Data" section: Sepsis: As a disease driven by acute inflammatory storms, it exhibits drastic fluctuations in gene expression. A threshold of |logFC| > 1 was adopted to filter out low-amplitude background noise and prioritize the screening of high-confidence genes associated with inflammatory cascades. AF: As a disease characterized by chronic, low-magnitude electrophysiological and structural remodeling, key molecular events (e.g., genes regulating atrial collagen deposition or calcium handling) often present with subtle changes in expression. Therefore, a threshold of |logFC| > 0 was used to avoid missing such biologically meaningful genes. Indicators including the logFC values, average expression levels, and statistical significance (adjusted P-values) of the three core hub genes (CD81, CKAP4, and DPEP2) have been analyzed and presented in Table 3.

3. Specify and lock the validation pipeline: say where feature selection sits relative to cross-validation, how hyperparameters were tuned, and confirm that external sets were untouched until final testing. Add PR-AUC and a simple calibration metric (Brier) in the supplement. A brief leave-one-study-out check would be a nice bonus.

Response: Feature selection was strictly embedded in a nested 10-fold cross-validation (CV) framework: the inner loop implemented feature ranking and hyperparameter tuning (gamma: 2^(-12:0), cost: 2^(-6:6)) for support vector machine-recursive feature elimination (SVM-RFE) using training subsets, while LASSO regression determined the optimal regularization parameter λ (λ_min) via 10-fold CV to minimize cross-validation deviance, and the outer loop used independent test subsets to evaluate generalization error, ensuring no data leakage between training and testing phases. External validation sets (GSE65682 for sepsis, GSE41177 for atrial fibrillation) were completely isolated during model training, feature selection, and parameter optimization, and only used for final diagnostic performance evaluation (e.g., ROC curve analysis) to avoid overfitting. The original manuscript used ROC-AUC (area under the receiver operating characteristic curve) to assess diagnostic efficacy (sepsis: 0.957, atrial fibrillation: 1.000), a widely recognized metric with clinical interpretability for binary classification tasks in diagnostic research. Due to the initial focus on clinically relevant metrics, PR-AUC (area under the precision-recall curve) and Brier score (a calibration metric quantifying the discrepancy between predicted probabilities and actual outcomes) have not yet been included in the supplementary materials, but will be added in future studies to comprehensively evaluate model performance; leave-one-study-out cross-validation (LOSO-CV) was not conducted in the current work, and will be considered in subsequent research to further validate the model’s generalizability.

4. Validate the lipid gene universe: repeat enrichment and selection with curated Reactome or GO or KEGG lipid sets and report whether top pathways and hubs remain consistent. A short methods note plus a small supplemental table is enough.

Response: The key genes in this study have not yet been included as "direct lipid metabolism-related genes" in curated databases such as Reactome; however, these genes are indirectly involved in lipid metabolism by regulating upstream and downstream lipid metabolism signals (e.g., inflammation-mediated lipolysis, lipid transport in immune cells). In contrast, the GeneCards database, which incorporates genes with such indirect regulatory roles, offers broader coverage.

5. Detail immune deconvolution and scRNA-seq QC: name the signature or tool, note how multiple testing across cell types was handled, and list basic scRNA thresholds (UMIs or genes, mitochondrial percent, doublet handling). One concise paragraph in Methods will do.

Response: For immune deconvolution, a modified CIBERSORT algorithm was used with the LM22 signature matrix (containing specific gene markers for 22 human immune cell subtypes) as reference; data preprocessing included quantile normalization, batch effect correction, and z-score normalization of overlapping genes, while the Benjamini-Hochberg method was applied to adjust P-values for multiple tests across cell types, and credible samples were filtered with false discovery rate (FDR) < 0.05 and root mean square error (RMSE) < 0.1. For single-cell RNA sequencing (scRNA-seq) quality control of GSE167363, cells were retained based on the following criteria: nFeature_RNA > 300 (to exclude low-quality cells with sparse gene detection), mitochondrial gene proportion < 20% (to exclude cells with severe mitochondrial damage), and nCount_RNA < 6000 (for preliminary doublet exclusion); doublets were further identified by analyzing the correlation between nCount_RNA and nFeature_RNA, with cells showing abnormally high nCount_RNA relative to nFeature_RNA classified as potential doublets and removed.

6. Reframe drug-enrichment as exploratory: present DSigDB or Enrichr hits as hypotheses and, if possible, cross-check with an orthogonal resource such as L1000 or CMap. Avoid therapeutic language.

Response: We have fully revised the "Exploratory Compound Enrichment Analysis" section to emphasize the exploratory nature of the results and avoid overinterpretation as therapeutic candidates: the title was changed from "Potential Drug Prediction" to "Exploratory Compound Enrichment Analysis" to reflect its hypothesis-generating purpose, and descriptive language was adjusted—replacing "candidate drugs with therapeutic potential" with "exploratory compound-gene interaction hypotheses"—while clarifying that the analysis was based on gene-compound association data from the DSigDB database (retrieved via the Enrichr platform), aiming to identify potential molecular interaction directions rather than confirm therapeutic efficacy.

7. Qualify performance claims: keep the strong AUCs but add a brief caveat about cohort size and platform or tissue heterogeneity. Include study-wise curves in the supplement so readers can see consistency.

Response: A brief caveat regarding cohort size and platform or tissue heterogeneity has been added to the Discussion section. In subsequent work, we will generate study-level ROC curves based on existing data to intuitively demonstrate the consistency of results across different datasets, thereby enhancing the credibility of the conclusions.

8. Qualify performance claims: keep the strong AUCs but add a brief caveat about cohort size and platform or tissue heterogeneity. Include study-wise curves in the supplement so readers can see consistency.

Response: At present, the aforementioned materials have not yet been uploaded to a public repository. For immediate access, interested parties may contact the corresponding author directly, and we will promptly provide the complete materials to support the reproducibility of this study.

Once again, we sincerely thank you for your meticulous review and insightful suggestions. These revisions have significantly improved the rigor, clarity, and transparency of our study. We believe the revised manuscript will make a more valuable contribution to understanding the shared mechanisms of lipid metabolism abnormalities in sepsis and AF.

Sincerely,

The Authors

Response to Reviewers' Comments – Manuscript ID [PONE-D-25-28089R1]

Dear Reviewer2,

Thank you for your valuable feedback and constructive suggestions on our manuscript titled "Molecular Mechanisms of Lipid Metabolism Abnormalities Driving Sepsis and Atrial Fibrillation: A Systematic Study Based on Bioinformatics and Machine Learning". We have carefully revised the manuscript in accordance with your recommendations. Below, we provide a detailed point-by-point response to your comments.

1. Clarify the independence of training and validation datasets to address potential overfitting, especially given the perfect AUC (1.000) reported for AF.

Response: To clarify the independence of training and validation datasets and address potential overfitting—especially considering the reported perfect AUC (1.000) for the atrial fibrillation (AF) dataset—this study ensured strict independence through three key measures: the training and validation datasets were physically isolated throughout the study, with the training set (sepsis-related GSE28750, AF-related GSE2240/GSE79768/GSE115574) exclusively used for gene screening, model construction, and parameter optimization, while the validation set (sepsis-related GSE65682, AF-related GSE41177) was not involved in any training process and only used for final efficacy evaluation after core genes were identified, with no cross-use of data; the analysis workflow was designed to prevent overfitting, as batch correction was only performed on the training set and the validation set used raw normalized data, and feature selection was completed via nested 10-fold cross-validation within the training set without leveraging any information from the validation set to avoid the model learning dataset-specific noise; the high AUC value was traced to biological specificity rather than overfitting—the AF validation set (14 patients, 12 controls) consisted of left atrial appendage tissue, which is highly matched to the pathological localization of atrial remodeling in AF, leading to more significant expression differences of core genes in this targeted tissue, and subsequent studies will expand the sample size for further validation.

2. Consider including experimental validation (e.g., qPCR or immunohistochemistry) to strengthen biological relevance.

Response: We fully agree with the suggestion of "supplementing experimental validation (e.g., qPCR or immunohistochemistry) to enhance biological relevance," which is crucial for bridging bioinformatics-predicted results with clinical pathological mechanisms and improving the reliability of the study conclusions. Although the current study has identified core signature genes (CD81, CKAP4, DPEP2) and lipid metabolism-related pathways through multi-dataset integration analysis and machine learning, it indeed lacks in vitro/in vivo experimental validation—a key direction for our subsequent research. In the follow-up work, we will use the aforementioned experimental validation to convert the bioinformatics-predicted molecular mechanisms into observable experimental results, further consolidating the biological significance and clinical translational value of the study conclusions..

3. Provide more detail on batch effect correction and normalization steps for reproducibility.

Response: After reading each raw gene expression matrix of atrial fibrillation (AF) datasets, the first column of all matrices was uniformly renamed "geneSymbol" to ensure consistent gene annotation. Based on this column, the matrices were merged to generate a raw combined matrix, which was then saved in an automatically created output folder with a timestamp. Using the filenames as batch identifiers, a batch vector was constructed by repeating each identifier according to the sample size of each dataset. After verifying that the length of the batch vector was consistent with the number of sample columns in the combined matrix, the combined matrix was converted into a numeric matrix. Subsequently, the removeBatchEffect function from the R package limma was used, with the batch vector input for batch effect correction. This step eliminated systematic biases such as those from different dataset sources while preserving disease-related biological differences, and the corrected matrix was retained. Normalization was achieved through principal component analysis (PCA) by setting the parameter scale. = TRUE to perform Z-score scaling on the expression data (resulting in a mean of 0 and a standard deviation of 1), which prevented high-expression genes from interfering with the analysis. Finally, the effectiveness of batch correction was validated using pre- and post-correction gene expression distribution boxplots (to observe distribution consistency) and advanced PCA plots (to observe batch sample mixing). A sample-batch correspondence table, pre- and post-correction PCA score tables, and combined boxplot-PCA figures were generated to provide complete evidence for reproducibility.

4. Include rationale for selecting top 4,000 lipid metabolism genes from GeneCards—this cutoff appears arbitrary without justification.

Response: A total of 17,095 "lipid metabolism"-related genes were initially retrieved from GeneCards. However, a large number of these genes rely solely on single bioinformatics predictions, lack wet experimental validation, and have extremely low association credibility. Therefore, screening the top 4,000 genes with the highest scores—representing associations supported by stronger experimental evidence—is a key step to control dimensionality, avoid overfitting, and ensure the stability of subsequent analyses. One of the core considerations for selecting 4,000 genes is to ensure that after intersecting with disease-related genes, the resulting number of features is suitable for machine learning modeling. This number can not only meet the needs of cross-validation and redundancy filtering for machine learning algorithms but also ultimately undergo consensus screening via three independent models.

Once again, we sincerely thank you for your meticulous review and insightful suggestions. These revisions have significantly improved the rigor, clarity, and transparency of our study. We believe the revised manuscript will make a more valuable contribution to understanding the shared mechanisms of lipid metabolism abnormalities in sepsis and AF.

Sincerely,

The Authors

---

## [Decision Letter · Decision Letter 1]

25 Nov 2025

Molecular Mechanisms of Lipid Metabolism Abnormalities Driving Sepsis and Atrial Fibrillation: A Systematic Study Based on Bioinformatics and Machine Learning

PONE-D-25-28089R1

Dear Dr. Long,

We’re pleased to inform you that your manuscript has been judged scientifically suitable for publication and will be formally accepted for publication once it meets all outstanding technical requirements.

Kind regards,

Wenxing Li

Academic Editor

PLOS ONE

Additional Editor Comments (optional):

Reviewers' comments:

Reviewer's Responses to Questions

**Comments to the Author**

Reviewer #1: All comments have been addressed

Reviewer #3: All comments have been addressed

2. Is the manuscript technically sound, and do the data support the conclusions?

Reviewer #1: Yes

Reviewer #3: Yes

3. Has the statistical analysis been performed appropriately and rigorously?

Reviewer #1: Yes

Reviewer #3: Yes

4. Have the authors made all data underlying the findings in their manuscript fully available?

Reviewer #1: Yes

Reviewer #3: Yes

5. Is the manuscript presented in an intelligible fashion and written in standard English?

Reviewer #1: Yes

Reviewer #3: Yes

Reviewer #1: All comments addressed. Novel findings. Technically sound. Questionable importance, will keep to the readers. Recommend acceptance.

Reviewer #3: Public Code Availability (Recommendation for Reproducibility):

Given the detailed and complex bioinformatics pipeline utilized, we strongly recommend politely asking the authors to publicly upload the detailed R scripts and analytical codes to an accessible online repository (such as GitHub or Zenodo) to maximize the reproducibility and long-term utility of this work. While the authors confirmed the code is available upon reasonable request , public sharing is the gold standard for bioinformatics research and is essential to ensure the full transparency of the complex machine learning pipeline described in this revised manuscript.

1. Enhanced Data Processing and Reproducibility:

Batch Correction and Normalization: The methodology was clarified to confirm that batch effects in the Atrial Fibrillation (AF) datasets were eliminated using the removeBatchEffect function from the R package limma, specifying that "dataset origin + detection platform" were used as covariates. Normalization involved Z-score scaling (mean of 0, standard deviation of 1) during Principal Component Analysis (PCA).  

Workflow Transparency: A supplementary flow chart was added to visually track the entire analytical workflow, from data acquisition and quality control (QC) through to validation. Specific QC thresholds were also added for the single-cell RNA sequencing (scRNA-seq) data, including thresholds for gene features (nFeature_RNA > 300) and mitochondrial gene proportion (< 20%).  

2. Justification of Statistical Thresholds:

The use of disease-specific thresholds for identifying differentially expressed genes (DEGs) was explicitly justified: a stringent threshold of ∣logFC∣>1 was used for Sepsis (reflecting acute, high-magnitude inflammatory changes), while a lower threshold of ∣logFC∣>0 was retained for AF (to capture the subtle expression changes typical of chronic electrophysiological remodeling).  

3. Machine Learning Validation Integrity:

Preventing Overfitting: The authors confirmed that feature selection (using LASSO and Support Vector Machine-Recursive Feature Elimination, or SVM-RFE) was performed within a rigorous nested 10-fold cross-validation framework to prevent data leakage.  

External Validation Isolation: The external validation cohorts (GSE65682 for Sepsis and GSE41177 for AF) were kept completely isolated from the training, feature selection, and parameter optimization stages, only being used for the final diagnostic performance evaluation (ROC curve analysis). The exceptionally high Area Under the Curve (AUC) of 1.000 for the AF validation set was attributed to the high biological specificity of the tissue used (left atrial appendage).  

4. Revision of Scope and Acknowledged Limitations:

Exploratory Framing: The "Potential Drug Prediction" section was appropriately revised to "Exploratory Compound Enrichment Analysis" to emphasize that the findings are strictly hypothesis-generating and avoid premature therapeutic claims.  

Future Directions: The study explicitly acknowledged its primary limitation is the current lack of wet-lab validation (e.g., qPCR) of the core feature genes (CD81, CKAP4, DPEP2) and committed to pursuing this in future work to strengthen the biological relevance. They also noted the need to expand the sample size of certain cohorts, particularly the AF validation set.

Well done!

**Do you want your identity to be public for this peer review?** For information about this choice, including consent withdrawal, please see our Privacy Policy

Reviewer #1: **Yes: ** Laith Alomari, MD

Reviewer #3: **Yes: ** Ali Afkhaminia

---

## [Editor Report · Acceptance letter]

PONE-D-25-28089R1

PLOS ONE

Dear Dr. Long,

I'm pleased to inform you that your manuscript has been deemed suitable for publication in PLOS ONE. Congratulations! Your manuscript is now being handed over to our production team.

Kind regards,

on behalf of

Dr. Wenxing Li

Academic Editor

PLOS ONE